# Outcome after acute ischemic stroke is linked to sex-specific lesion patterns

Anna K. Bonkhoff [1✉], Markus D. Schirmer[1,2], Martin Bretzner [1,3], Sungmin Hong[1], Robert W. Regenhardt [1], Mikael Brudfors[4], Kathleen L. Donahue[1], Marco J. Nardin[1], Adrian V. Dalca[5,6], Anne-Katrin Giese[7], Mark R. Etherton[1], Brandon L. Hancock[6], Steven J. T. Mocking[6], Elissa C. McIntosh[6], John Attia [8,9], Oscar R. Benavente[10], Stephen Bevan[11], John W. Cole[12], Amanda Donatti[13], Christoph J. Griessenauer[14,15], Laura Heitsch[16,17], Lukas Holmegaard[18,19], Katarina Jood[18,19], Jordi Jimenez-Conde[20], Steven J. Kittner[12], Robin Lemmens[21,22], Christopher R. Levi[23,24], Caitrin W. McDonough[25], James F. Meschia [26], Chia-Ling Phuah[17], Arndt Rolfs[27], Stefan Ropele [28], Jonathan Rosand[1,6,29], Jaume Roquer[20], Tatjana Rundek[30], Ralph L. Sacco[30], Reinhold Schmidt[28], Pankaj Sharma[31,32], Agnieszka Slowik[33], Martin Söderholm[34,35], Alessandro Sousa[13], Tara M. Stanne[36], Daniel Strbian [37], Turgut Tatlisumak[38,39], Vincent Thijs[40,41], Achala Vagal[42], Johan Wasselius[43,44], Daniel Woo[45], Ramin Zand[46], Patrick F. McArdle[47], Bradford B. Worrall [48], Christina Jern[36,49], Arne G. Lindgren[50,51], Jane Maguire[52], Danilo Bzdok [53,54], Ona Wu[6], MRI-GENIE and GISCOME Investigators and the International Stroke Genetics Consortium* & Natalia S. Rost[1]

Acute ischemic stroke affects men and women differently. In particular, women are often reported to experience higher acute stroke severity than men. We derived a low-dimensional representation of anatomical stroke lesions and designed a Bayesian hierarchical modeling framework tailored to estimate possible sex differences in lesion patterns linked to acute stroke severity (National Institute of Health Stroke Scale). This framework was developed in 555 patients (38% female). Findings were validated in an independent cohort (n = 503, 41% female). Here, we show brain lesions in regions subserving motor and language functions help explain stroke severity in both men and women, however more widespread lesion patterns are relevant in female patients. Higher stroke severity in women, but not men, is associated with left hemisphere lesions in the vicinity of the posterior circulation. Our results suggest there are sex-specific functional cerebral asymmetries that may be important for future investigations of sex-stratified approaches to management of acute ischemic stroke.

---

A full list of author affiliations appears at the end of the paper.

Stroke affects >15 million people each year[1]. It is known to result in a substantial overall degree of long-term impairment across men and women[2,3]. However, numerous epidemiological studies indicate clinically relevant, sex-related differences in the characteristics of ischemic cerebrovascular disease[4,5]. For instance, due to a longer life expectancy, more women than men experience a stroke each year[6]. Expected demographical changes, i.e., an aging population, will widen this gap further: in the US, projections suggest that ~200,000 more women will be disabled after stroke than men by 2030[7].

Further sex differences relate to women more often presenting with non-classic stroke symptoms, such as fatigue or changes in mental status[8,9], and having a higher risk of delays in hospital arrival[10,11]. Also, women feature a higher risk of cardioembolic stroke due to atrial fibrillation[12], which may contribute to the often-observed higher acute ischemic stroke (AIS) severity in female patients[13]. This excess in stroke severity in women persists even after adjusting for their greater age at onset, comorbidities, and prestroke level of independence[14,15]. Importantly, women seem to experience more severe strokes despite comparable lesion sizes in men and women[16]. In fact, a similar observation of sex-specific lesion volume effects was noted in the case of aphasia, where women had a smaller lesion volume threshold to cause aphasia than men[17].

Going beyond lesion volume, lesion-symptom mapping studies have enriched our understanding of anatomically unique lesion locations underlying specific symptoms post-stroke[18–20]. In the case of stroke severity, these analyses have determined widespread lesions in white matter, basal ganglia, pre- and postcentral gyri, opercular, insular, and inferior frontal regions to be most relevant for a higher stroke severity, especially if affecting the left hemisphere[21]. While these lesion-symptom studies have uncovered eloquent lesion locations with high spatial resolution, they have been systematically blind to any potential sex disparities. If considered at all, sex was treated as a nuisance variable and regressed out prior to the main analysis[21]. Thus, none of the recently employed analytical approaches in clinical neuroimaging allowed for a dedicated, explicit investigation of sex-specific lesion pattern effects in relation to continuous outcome scores.

In this work, we aim to design and conduct a lesion-symptom analysis capable of capturing male- and female-specific lesion patterns, underlying stroke severity in a statistically robust and spatially precise manner to address previous methodological constraints. For this purpose, we leverage neuroimaging data originating from two large, independent hospital-based cohorts gathering data of 555 (derivation) and 503 (validation) AIS patients in total. We tailor and deploy sex-aware hierarchical Bayesian models to simulate predictions of AIS severity and to elucidate the sex-specific effects of lesion patterns affecting similar brain regions in women and men. We seek to map the lesion constellations underpinning female-specific more severe strokes, potentially indicating sex-specific maps of functional deficits on the one hand and encouraging more sex-aware acute stroke treatment decisions on the other. Such a sex-informed acute stroke care has the potential to alleviate the burden of disease on an individual patient level, as well as broader and socioeconomically relevant levels.

## Results

We here present a generative analysis of acute stroke severity, putting a particular focus on sex-specific lesion pattern effects. We successively combined (1) the automated low-dimensional embedding of high-dimensional DWI-derived lesion information via non-negative matrix factorization (NMF)[22], and (2) probabilistic modeling, based on the latent NMF embedding, to simulate the prediction of acute stroke severity, as measured by the National Institute of Health Stroke Scale (NIHSS)[23]. We thus first determined pivotal, general lesion pattern effects across all patients and successively concentrated on similarities and differences between men and women (sex assessed by patients' medical records). We interpreted explanatory relevances on the level of NMF-derived low-dimensional lesion representations, that we call lesion atoms, as well as the same relevances transformed back to the level of the anatomical gray matter brain regions and white matter tracts.

**Stroke sample characteristics.** The derivation cohort consisted of 208 female and 347 male patients ($n = 555$ in total, mean age (standard deviation (SD)): 65.0(14.8); 38% women, as indicated by patients' medical records). The main outcome of interest was the acute NIHSS-based stroke severity score within the first 48 h after admission (median(interquartile range): 3(6), Supplementary Fig. 1). More than one-fourth of stroke patients had a history of hypertension (28.1%), 19.5% had a diagnosis of diabetes mellitus, 6.3% atrial fibrillation, and 7.6% coronary artery disease. As expected based on prior reports[12], women showed a higher frequency of atrial fibrillation than men (9.1% vs. 4.6%, $p = 0.05$; c.f., Table 1 for further sex-specific numbers). Acute stroke lesion

### Table 1 Patient characteristics.

| | All participants (n = 555) | Women (n = 208) | Men (n = 347) | Statistical comparison of male and female patients |
|---|---|---|---|---|
| Age | 65.0 (14.8) | 67.7 (16.3) | 63.3 (13.5) | $p = 0.001$* |
| Sex | 62% male, 38% female | — | — | — |
| NIHSS | 5.0 (5.9) (median(iqr): 3 (6)) | 5.6 (6.6) (median (iqr): 3(6)) | 4.7 (5.5) (median (iqr): 3(5)) | $p = 0.09$ |
| Normalized DWI-derived stroke lesion volume (ml) | 13.7 (29.9) (median(iqr): 1.7(11.6)) | 13.6 (31.8) (median (iqr): 1.5(9.7)) | 13.7 (28.7) (median (iqr): 1.7(13.2)) | $p = 0.98$ |
| White matter hyperintensity lesion volume (ml) | 11.5 (13.5) | 12.1 (13.3) | 11.1 (13.6) | $p = 0.42$ |
| Hypertension | 28.1% | 29.3% | 27.4% | $p = 0.63$ |
| Diabetes mellitus type 2 | 19.5% | 17.8% | 20.5% | $p = 0.51$ |
| Atrial fibrillation | 6.3% | 9.1% | 4.6% | $p = 0.05$* |
| Coronary artery disease | 7.6% | 6.7% | 8.1% | $p = 0.62$ |

Mean (SD) unless otherwise noted. The groups of male and female patients were compared via two-sample $t$ tests or two-sided Fisher's exact test as appropriate. Asterisks indicate significant differences between men and women. The disproportionate representation of men and women may reflect an undersampling of female patients as frequently observed in randomized clinical stroke trials[87], and may largely stem from the noninclusion of elderly and more severely affected female patients.

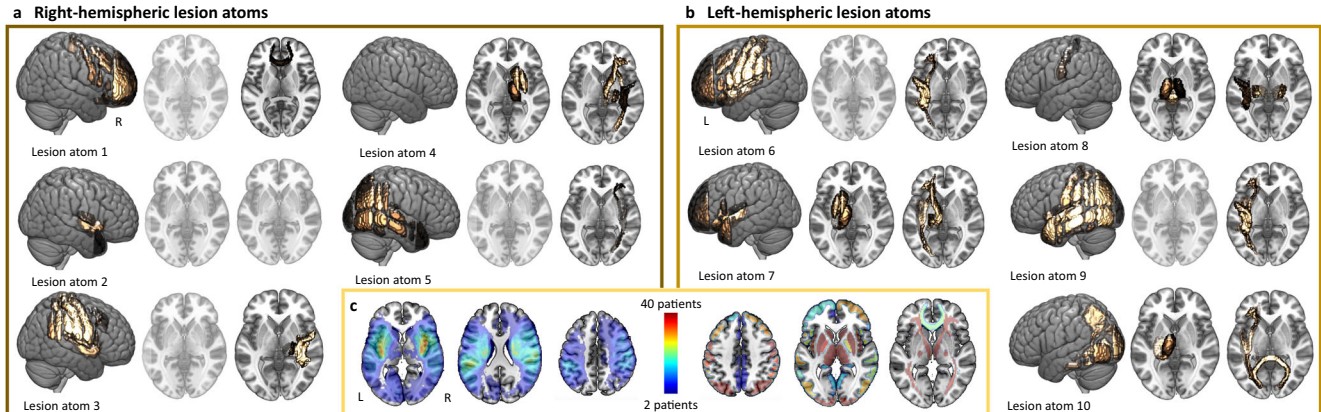

**Fig. 1 Archetypical stroke patterns, lesion atoms, as resulting from non-negative matrix factorization-based dimensionality reduction.** A data-driven pattern discovery framework enabled the derivation of coherent patterns of stroke lesion topographies directly from the segmented high-resolution brain scans from 555 stroke patients. This unsupervised, multi-to-multi mapping approach led to unique, predominantly right-hemispheric (**a**) or left-hemispheric (**b**) lesion patterns. In case of either one hemisphere, individual lesion atoms represented anatomically coherent cortical and subcortical brain regions and respective white matter tracts, and had varying emphases on more anterior, medial, and posterior regions. While subcortical basal ganglia lesions and cortical lesions in anterior and insular regions were combined in a single lesion atom on the left side of the brain, these patterns were characterized by two separate lesion atoms on the right side of the brain. Lesion atom 8 comprised several brain regions in the left hemisphere; however, also comprised the brainstem and, to a lesser degree, the right thalamus. This pattern likely arose since we did not exclude patients with bilateral stroke ($n = 21$, 38% female). Some lesion atoms did not comprise any substantial contributions from subcortical brain regions and are shown in transparent. The anatomical plausibility of our derived lesion pattern may particularly stem from the positivity constraint of the non-negative matrix factorization algorithm, an advantageous quality that motivated our choice of dimensionality reduction technique. Conversely, alternative matrix factorization algorithms, such as principal component analysis, would have hampered a straightforward interpretation of lesion pattern effects by encoding individual lesions through more incomprehensible additions and subtractions of low-dimensional lesion pattern. **c** Similarity of lesion patterns across patients. A voxel-wise lesion overlap is visualized on the left-hand side, while the right-hand side presents region-wise frequencies, i.e., the number of how often a specific region was affected. The maximum lesion overlap was localized subcortically and in the proximity of insular regions in the left and right vascular supply territory of the middle cerebral artery. Significant region-wise differences in lesion loads and frequencies did neither arise between men and women nor between left and right hemispheres (Supplementary Data 1 and 2). Source data are provided as a Source data file.

volume did not differ significantly between men and women (two-sided $t$ test: $p = 0.98$). Moreover, neither the frequencies of how often each cortical and subcortical gray matter region or white matter tract was affected, nor the numbers of lesioned voxels within each region of interest differed significantly between the sex categories, or between the left and right hemisphere (all Fisher's exact tests or two-sided $t$ tests $p > 0.05$, Bonferroni-corrected for multiple comparisons, c.f., Fig. 1c, Supplementary Fig. 1, and Supplementary Data 1 and 2).

**Anatomy of the extracted lesion atoms in stroke patients.** We reduced the high-dimensional lesion voxel space by first computing the number of lesioned voxels within each of 109 cortical and subcortical gray matter regions, as well as 20 white matter tracts. Subsequently, we employed unsupervised NMF to ten final distinct lesion topographies, or lesion atoms. Derived low-dimensional lesion atoms were found to represent anatomically plausible, hemisphere-specific lesion patterns. The centers of these lesion patterns varied from anterior to posterior and subcortical to cortical regions, and were broadly similar between hemispheres (Fig. 1a, b). Subcortical and cortical lesion patterns were represented in separate lesion atoms in the right hemisphere, while they were captured in a joint lesion atom on the left. In correspondence to the primary distribution of individual lesions, most of these lesion atoms related to infarcts in the left and right MCA-supply territories and to a lesser extent to infarcts in the posterior circulation. The maximum lesion overlap was localized in left and right subcortical MCA and insular region (Fig. 1c).

The low-dimensional representation of lesion topographies served as input for fully probabilistic, hierarchical linear regression models to explain acute stroke severity: first, we

examined general effects across all patients, on the level of lesion atoms and on the level of individual anatomical brain regions. Successively, we refined analyses and integrated an additional hierarchy capturing sex-specific effects. We stratified for male and female sex status and scrutinized sex-specific effects of lesion atoms and anatomical brain regions. Of note, we corrected all of these analyses for the covariates age, sex, stroke lesion volume, white matter hyperintensity lesion volume, and relevant comorbidities (atrial fibrillation, hypertension, diabetes mellitus, and coronary artery disease). Notably, we included sex as a variable in the model to differentiate between sex differences in stroke severity that were dependent and independent of lesion patterns. If, for example, stroke severity was generally higher in women, without any link to the actual lesion pattern, conceivably due to a longer delay between symptom onset and hospital admission and decreased likelihood of acute treatment administration, this effect would be represented by the Bayesian posterior distribution of this sex variable, but not in the sex-specific lesion atom Bayesian posterior distribution. In contrast, sex-specific lesion atom distributions would indicate interaction effects on the outcome, i.e., effects that were specific to an individual's sex and the precise lesion atom. Lastly, it is important to note that this adjustment for global sex differences was independent of the exact knowledge or measurements of their causes, i.e., we did not need to include any information on the delay in hospital admission explicitly.

**Lesion atom and regional relevance for stroke severity.** Out of the ten derived lesion atoms, five atoms possessed a substantial explanatory relevance for acute stroke severity. This relevance was inferable from Bayesian posterior distributions of lesion atoms that did not substantially overlap with zero. In the right hemisphere, the most relevant lesion atom included subcortical

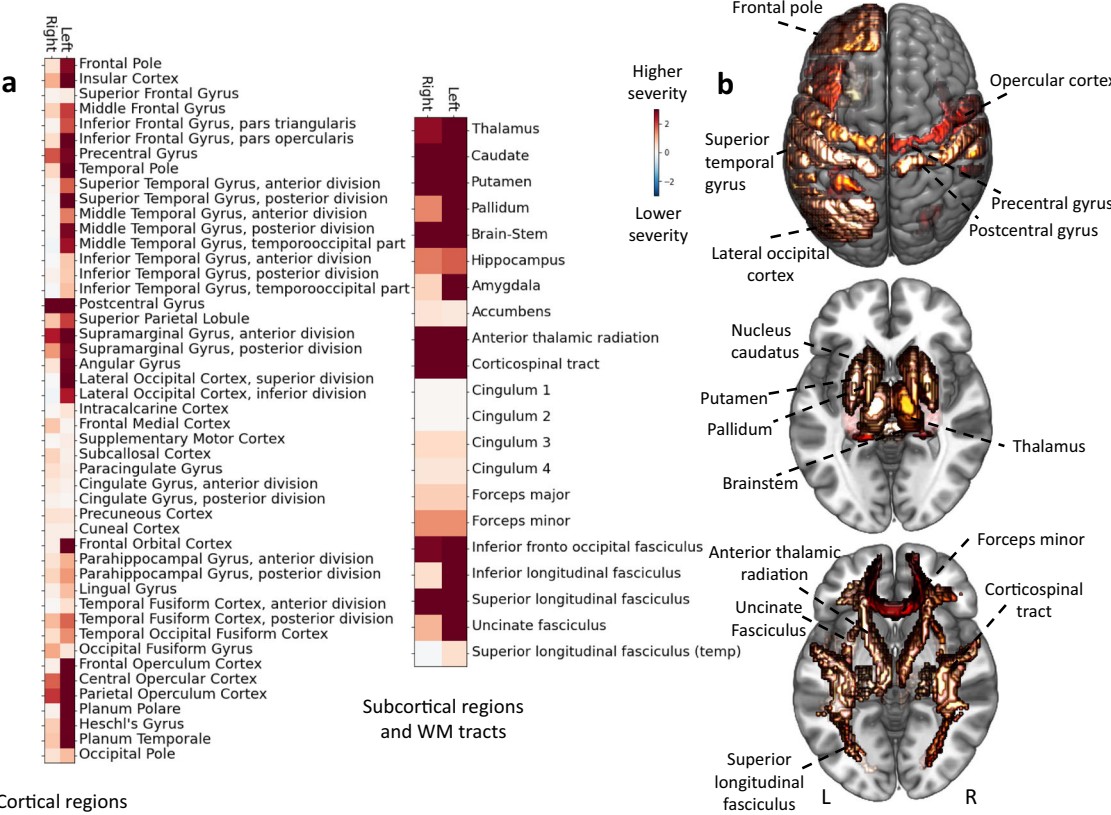

**Fig. 2 Local brain regions explaining NIHSS-based stroke severity across 555 patients. a** Relevant cortical and subcortical gray matter regions, as well as relevant white matter (WM) tracts. Shows collection of marginal Bayesian posterior distributions from the hierarchical model to explain high vs. low symptom severity (NIHSS). Lesions affecting pre- and postcentral gyri, as well as opercular regions of both hemispheres explained a higher stroke severity. Further brain-behavior effects were left-lateralized: multiple regions, including left middle and inferior frontal gyrus, as well as superior and middle temporal explained a higher stroke severity only when affecting the left hemisphere. While bilateral subcortical regions, in general, had substantial effects on stroke severity, the highest weights were assigned to the putamen and caudate, as well as anterior thalamic radiation, corticospinal tract, and inferior fronto-occipital fasciculus of the left hemispheres. **b** Brain renderings of region-wise relevances in the explanation of NIHSS-based stroke severity. Source data are provided as a Source data file.

regions, i.e., thalamus, nucleus caudatus, putamen, and globus pallidus (lesion atom 4: mean of the posterior distribution = 2.18, highest probability density interval (HPDI) of the posterior distribution covering 90% certainty = 1.43–2.99, Supplementary Fig. 2A). In the left hemisphere, the two most relevant lesion atoms were characterized by both subcortical and cortical regions (lesion atom 7: posterior mean = 3.76, HPDI = 2.99–4.49; lesion atom 8: posterior mean = 4.8, HPDI = 2.89–6.98, Supplementary Fig. 2B). Affected left and right subcortical regions were similar in explaining acute stroke severity, whereas left cortical affected regions additionally included inferior frontal, insular and superior temporal gyrus regions, as well as the postcentral gyrus.

Once projected back to the level of individual gray matter regions and white matter tracts, similarities and disparities between the left and right hemispheres became apparent as well. Subcortical regions, most notably thalamus, nucleus caudatus, putamen, globus pallidus, and several white matter tracts (anterior thalamic radiation, corticospinal tract, inferior fronto-occipital fasciculus, and superior longitudinal fasciculus) explained higher stroke severity, independent of the lesioned hemisphere (Fig. 2). Likewise, cortical pre- and postcentral, as well as supramarginal gyrus and parietal regions explained higher stroke severity in both the left and right hemispheres. In contrast, further cortical effects were more pronounced and more widespread in the left hemisphere. These enhanced left-sided effects mainly related to middle and inferior frontal gyri, as well as

superior and middle temporal gyri, insular cortex, and broader opercular regions.

In summary, we derived stroke severity-linked lesion patterns that highlighted the general importance of subcortical gray matter regions and white matter tracts, as well as of bilateral cortical motor regions, and additional left-lateralized cortical regions, likely underlying language function.

**Differences in lesions patterns between men and women.** Next, we concentrated on sex differences in eloquent lesion patterns. We refined our Bayesian model and introduced a hierarchical structure that allowed lesion atom effects on stroke severity to vary by sex. Previous findings suggest a higher stroke severity in women in general, the extent of which can neither be sufficiently explained by their advanced age nor differences in comorbidities, the prestroke level of independence or lesion volume[14–16].

Main patterns of explanatory relevances remained similar to the preceding analysis across all patients: for both men and women, the same three lesion atoms, that had already emerged in the joint analyses, had the highest explanatory relevance. The right-hemispheric lesion atom denoted subcortical regions, among others representing thalamus, nucleus caudatus, putamen, and globus pallidus (lesion atom 4: men: posterior mean = 1.89, HPDI = 1.00–2.90, women: posterior mean = 2.64, HPDI = 1.44–3.82, Fig. 3). The lesion atoms in the left hemisphere combined the

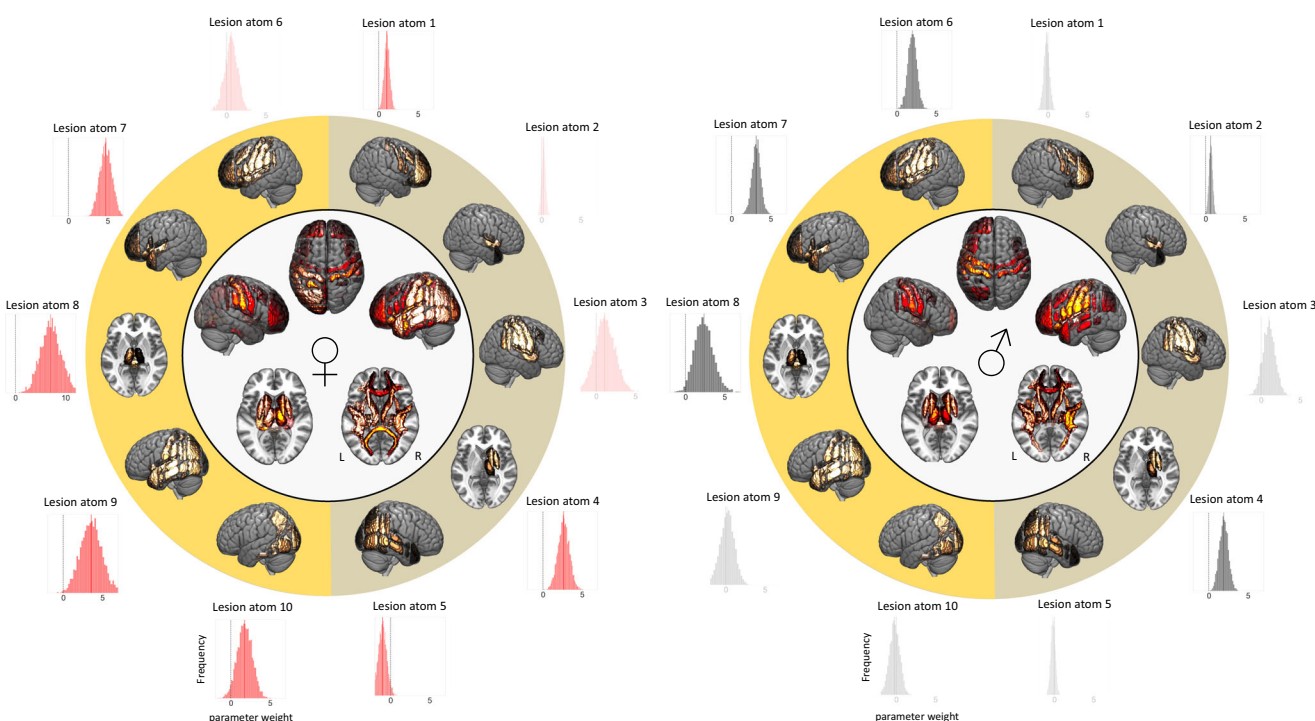

**Fig. 3 Sex-specific Bayesian posterior distributions of all ten lesion atoms and overall whole-brain region-wise relevance to explain stroke severity in women (left) and men (right).** Our Bayesian framework was purpose-designed to enable fully probabilistic estimations of the parameters that quantify the associations of the ten lesion atoms with stroke severity. These Bayesian posterior parameter distributions are shown in outer circles, corresponding lesion atom renderings are presented in the subjacent circle (right-hemispheric lesion atoms: shaded in yellow-olive, left-hemispheric lesion atoms: orange-yellow; distributions that substantially diverged from zero are nontransparent). Lesion atoms 7 and 8 of the left hemisphere and lesion atom 3 of the right hemisphere had the highest weights, implying a high relevance in explaining stroke severity, in both men and women. In view of seven relevant lesion atoms in women (lesion atoms: 1, 4, 5, 7–10), yet only five of such relevant lesion atoms in men (lesion atoms: 2, 4, 6–8), lesion patterns were more widespread in women compared to men. This female-specific more widespread pattern becomes additionally apparent in whole-brain renderings of region-wise relevances explaining stroke severity, as visualized in circle centers (c.f., Fig. 4 for details). Source data are provided as a Source data file.

same subcortical structures, as well as the brainstem and additional cortical regions, all of them in proximity to the insular cortex in the left hemisphere (lesion atom 7: men: posterior mean = 3.15, HPDI = 2.22–4.02, women: posterior mean = 4.76, HPDI = 3.49–6.07; lesion atom 8: men: posterior mean = 2.37, HPDI = 0.393–4.27, women: posterior mean = 7.04, HPDI = 3.75–10.50, Fig. 3).

Women presented with generally more widespread explanatory relevances as seven out of ten lesion atoms posterior distributions did not overlap with zero (Fig. 3, lesion atoms 1, 4, 5, 7–10). In men, only five lesion atoms posterior distributions did not overlap with zero (Fig. 3, lesion atoms 2, 4, 6–8).

Once projected back to the level of individual brain regions, these more widespread lesion pattern effects in women were also visible, in particular regarding cortical gray matter regions (c.f., Figs. 3 and 4). Manifest differences between men and women emerged for five specific lesion atoms: for women, the right-hemispheric lesion atom 1 and the four left-hemispheric lesion atoms 7–10 had substantially higher explanatory relevances, i.e., the distribution of the difference between posterior distributions of male and female patients did not overlap with zero (Fig. 5). Lesion atom 1 was mainly characterized by right-sided frontal, insular and opercular, as well as precentral regions. Lesion atoms 7–10 represented cortical and subcortical regions of the entire left hemisphere.

**Validation analyses**. Similar main findings were shown when repeating the analyses in an independent, multisite dataset[24]. Again, we extracted ten lesion atoms that captured typical stroke patterns in the left and right hemispheres in low dimensions (Supplementary Fig. 3). While there were subtle differences of

lesion embeddings, likely expressing sample-specific lesion distributions, lesion atoms of the derivation and validation dataset were overall highly correlated. This high correlation indicated that our unsupervised approach facilitated the derivation of similar lesion topography embeddings in both independent datasets (Supplementary Table 2). In addition, back-projected region-wise relevances were highly correlated when computed based on the other cohort's lesion embedding (derivation cohort: $r = 0.84$, $p < 0.001$; validation cohort: $r = 0.78$, $p < 0.001$). Both of these correlation analyses combined, thus highlighted the distillation of largely similar archetypical lesion pattern in both cohorts and the independence of results from the concrete lesion embedding.

The most relevant regions explaining stroke severity were also located subcortically in the left and right hemisphere, as well as in bilateral precentral and postcentral gyri and left-hemispheric insular and opercular regions (Supplementary Fig. 4a). Similar to our analysis before, women presented more widespread eloquent lesion patterns compared to men (Supplementary Fig. 4b, c). In particular, we found substantial differences between men and women in a lesion atom that predominantly comprised left-hemispheric, presumably posterior cerebral artery-supplied regions (lesion atom 10: difference distribution: mean = −2.68, HPDI = −4.92 to −0.733 (i.e., no overlap with zero)). Sex differences relating to this specific left-sided lesion atom thus appeared to be the most pronounced and robust.

**Ancillary analyses**. We aimed to gain further insights into the influences of (i) the exact time of imaging and stroke severity

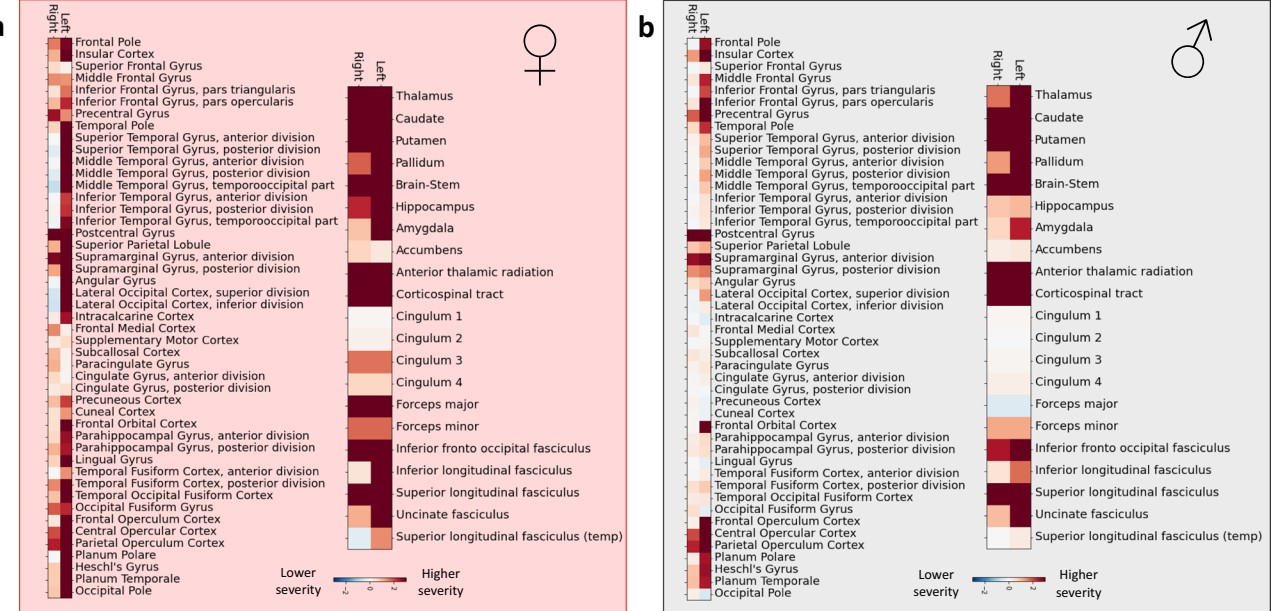

**Fig. 4 Local brain regions explaining stroke severity. a** Female-specific results (208 women) and **b** male-specific results (347 men). In both men and women, subcortical lesions affecting gray matter regions and white matter tracts explained higher stroke severity. Similarly, cortical presumptive bilateral motor and left-lateralized language regions (e.g., especially bilateral precentral and postcentral gyri, left-sided inferior frontal and superior, middle temporal gyri) also explained higher stroke severity. In difference to men, women featured more widespread and also more pronounced lesion pattern, including a greater range of cortical regions contributing to stroke severity, e.g., the left superior, middle and inferior temporal gyrus, left angular gyrus and lateral occipital cortex, lingual gyrus, as well as precuneus and parahippocampal cortex gyrus. These differences in eloquent lesion patterns arose despite comparable total lesion volumes for men and women (two-sided t test: p > 0.05). Source data are provided as a Source data file.

acquisition, (ii) cardioembolic vs. non-cardioembolic stroke subtypes, and (iii) potential hormonal effects.

While data were acquired within the first 48 h after admission for the entire derivation cohort, we aimed to reduce potential confounds due to varying times of imaging and stroke severity acquisition, as well as any acute revascularization therapy effects by specifically investigating those patients with MRI imaging and NIHSS score available upon their admission. These criteria were fulfilled by 152 patients in total (men: 87, mean age(SD): 63.1 (12.3), mean NIHSS(SD): 4.6(5.6); women: 65, mean age(SD): 69.6(14.5), mean NIHSS(SD): 5.5(6.5); two-sided $t$ tests: age: $p = 0.003$, NIHSS: $p = 0.45$, Supplementary Fig. 6). The Bayesian posterior difference distribution of lesion atom 10, representing presumed left posterior cerebral artery (PCA)-supplied brain regions, indicated a substantial female-specific effect in explaining a higher stroke severity (Supplementary Fig. 7).

Women, especially if older, are known to more frequently experience atrial fibrillation and cardioembolic strokes[12]. The motivation of our second ancillary analysis was to investigate whether observed sex differences in explanatory lesion patterns merely stemmed from varying frequencies of cardioembolic strokes. As information on stroke subtype was available for subgroups of each cohort only, we here pooled data from the derivation and validation cohort to rest analyses on a sample as large as possible. Approximately one-fourth of the 880 patients in total experienced a cardioembolic stroke (203 patients with cardioembolic stroke: men: 108, mean age(SD): 66.6(14.3), mean NIHSS(SD): 6.3(5.4), women: 95, mean age(SD): 74.3(12.9), mean NIHSS(SD): 7.5(7.0); two-sided $t$ tests: age: $p < 0.001$, NIHSS: $p = 0.17$; 677 patients with non-cardioembolic stroke: men: 419, mean age(SD): 62.8(13.3), mean NIHSS(SD): 4.7(5.3), women: 258, mean age(SD): 63.7(16.4), mean NIHSS(SD): 5.1(5.7); two-sided $t$ tests: age: $p = 0.45$, NIHSS: $p = 0.44$, Supplementary Fig. 8). As expected, women featured a higher frequency of cardioembolic strokes (Fisher's exact test: $p = 0.03$). We detected substantial

female-specific lesion atom effects independent of whether comparing men and women with cardioembolic strokes, or men and women with non-cardioembolic stroke genesis: in case of cardioembolic stroke, difference distributions of lesion atoms 4, 6, and 8 suggested a higher explanatory relevance exclusively in women (Supplementary Fig. 9), while lesion atoms 1, 2, 9, and 10 had a substantially higher relevance in women in case of non-cardioembolic stroke (Supplementary Fig. 10). Since we extracted female-specific effects in strata of both only cardioembolic and non-cardioembolic stroke patients, these results rendered the interpretation of lesion pattern effect differences due to varying frequencies of cardioembolic stroke unlikely. Furthermore, all of these female-specific effects emerged for samples that were fairly even in their numbers of men and women (c.f., cardioembolic stroke: 108 men and 95 women), or did not comprise any significant differences in age and stroke severity (c.f., non-cardioembolic stroke: two-sided $t$ tests: age: $p = 0.45$, NIHSS: $p = 0.44$), which additionally increased the confidence that sex differences did not artificially arise from these differences.

Finally, we aimed to explore the potential effects of sex hormones, such as estrogen, which are known to be markedly affected by menopause[25]. We stratified the entire group of patients according to their sex and an age cutoff of 52 years, the median age at menopause[26]. All of the female-specific lesion atom effects, as observed in the main analysis, disappeared in the analysis of all men and women below the age of 52 years (men: 113, mean age(SD): 43.1(8.5), mean NIHSS(SD): 5.4(6.1), women: 87, mean age(SD): 42.1(7.9), mean NIHSS(SD): 4.3(5.2); two-sided $t$ tests: age: $p = 0.41$, NIHSS: $p = 0.19$; Supplementary Fig. 11). What is more, lesion atom 7 was now assigned a higher relevance in male patients (Supplementary Fig. 12). In contrast, we observed female-specific higher relevances in three lesion atoms (lesion atoms 1, 9, and 10), when comparing men and women in the subgroup of patients above the age of 52 years of

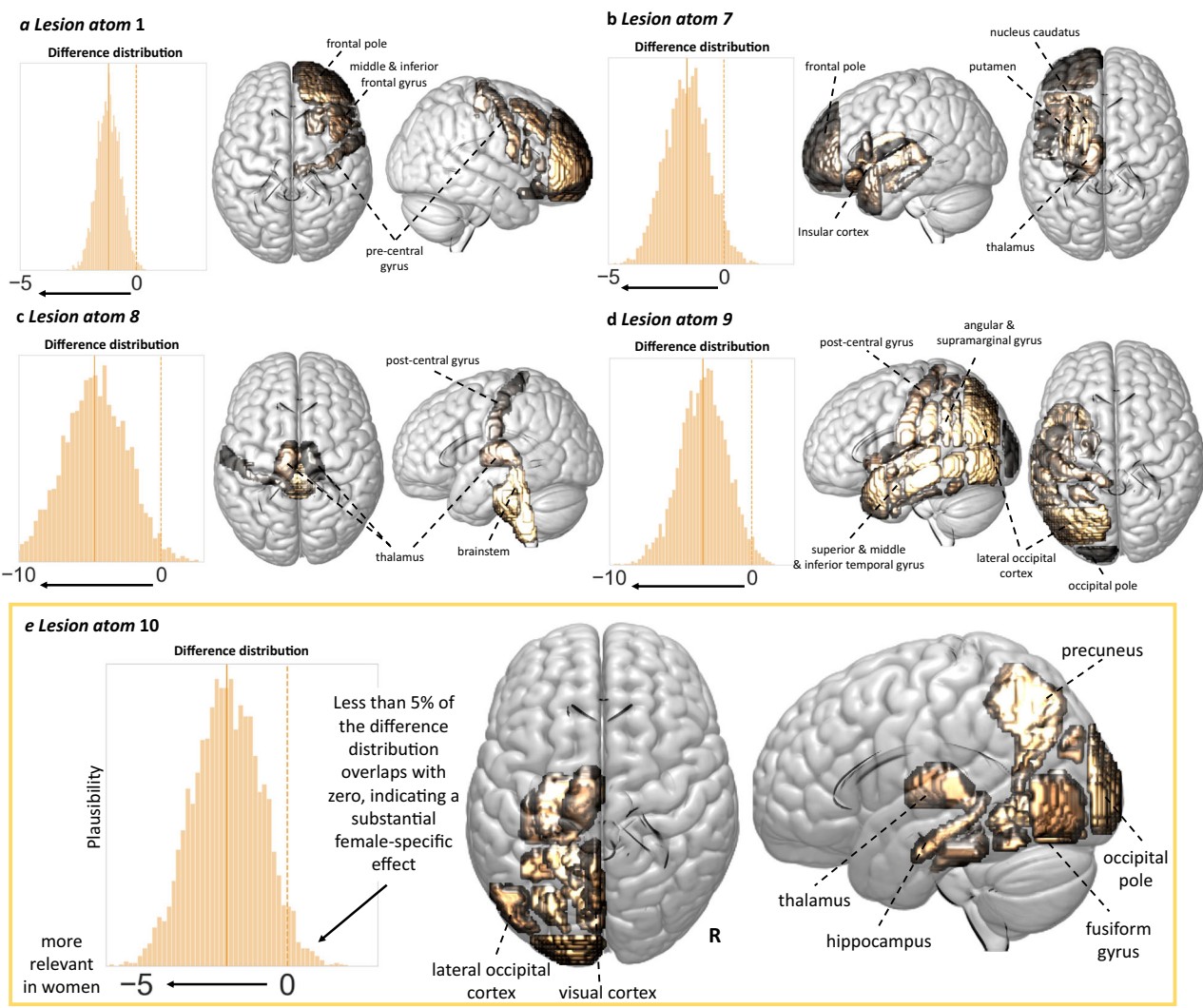

**Fig. 5 Five lesion atoms showed substantially higher relevance for explaining stroke-induced symptom severity in women than men. a** Lesion atom 1. Less than 5% of samples of the difference Bayesian posterior distribution between men and women (i.e., posterior distribution of lesion atom 1 in men—posterior distribution of lesion atom 1 in women) overlapped with zero, suggesting a substantially larger lesion atom 1 effect in women. This specific lesion pattern represented right-hemispheric lesions in frontal, insular, and precentral regions (difference distribution: mean = −1.2, HPDI = −1.96 to −0.31). **b–d** Lesion atoms 7–9. Almost all lesion atoms of the left hemisphere, but the one representing precentral cortex and middle and inferior frontal gyrus regions, indicated more pronounced effects on stroke severity in women. Lesion atom 7 highlighted left-sided insular cortex and subcortical regions (difference distribution: mean = −1.61, HPDI = −3.12 to −0.02). Lesion atom 8 comprised brainstem lesions, as well bilateral, left-hemispherically more distinct thalamus lesions (difference distribution: mean = −4.67, HPDI = −8.55 to −1.08) and lesion atom 9 widespread left-sided cortical lesions relating to the superior, middle and inferior temporal gyri, postcentral cortex, angular and supramarginal gyrus, and latero-occipital cortex (difference distribution: mean = −3.42, HPDI = −6.15 to −0.63). **e** Lesion atom 10. Lesions in left-sided brain regions of the presumed posterior circulation were associated to more severe strokes specifically in women (difference distribution: mean = −1.99, HPDI = −3.90 to −0.22). The sex difference for lesion atom 10 can be considered the most pronounced and robust one, given it was reliably observable in all ancillary analyses and was replicated in an independent dataset of stroke patients. All analyses were corrected for overall sex effects, i.e., lesion atom-independent effects (e.g., due to potential female-specific delayed hospital arrival). Of note, this correction was independent of the explicit knowledge or measurement of the causes of these global sex differences. Source data are provided as a Source data file.

age (Supplementary Fig. 13, men: 533, mean age(SD): 68.4(9.5), mean NIHSS(SD): 4.9(5.2), women: 325, mean age(SD): 73.0 (11.0), mean NIHSS(SD): 6.0(6.3); two-sided *t* tests: age: *p* < 0.001, NIHSS: *p* = 0.004). Overall, female-specific effects were thus noticeably more pronounced with advanced age. Given that we ascertained female-specific effects for both cardioembolic, as well as non-cardioembolic stroke subtypes in the previous ancillary analysis, the observations for older female patients are unlikely due to an increased frequency of cardioembolic stroke and linked lesion patterns. Further, this observable sex–age interaction might also explain why we discovered more extensive female-specific

effects in the derivation cohort, featuring significantly older female patients, than in the validation cohort that was characterized by a nonsignificant age difference between men and women.

Of note, female-specific effects relating to lesion atom 10, representing left-hemispheric presumably PCA-supplied regions, were the only ones that consistently emerged in all of the three ancillary analyses. Thus, these results confirmed the notion arising from the main derivation and validation analyses that sex differences concerning this lesion atom were the most reliable ones.

## Discussion

In this study, we combined a probabilistic lesion-symptom mapping technique with empirical lesion data originating from two large independent cohorts of 555 (derivation) and 503 (validation) AIS patients to derive and validate sex-specific lesion patterns underlying NIHSS-based acute stroke severity.

Across all patients, lesion patterns highlighted the relevance of bilateral subcortical and white matter regions, as well as bilateral cortical motor regions and left-lateralized cortical insular and opercular regions, likely representing regions underlying language function. This distribution of main weights was rendered particularly plausible, given that the NIHSS scale assigns the majority of its points to motor and language functions. In addition, results from both of our cohorts, and those results originating from established voxel-wise lesion-symptom mapping (VLSM) analyses[21] were very similar. This congruency increased the confidence in our methodological approach, as well as in the accuracy and reproducibility of results.

When comparing men and women, eloquent lesion patterns were generally more widespread in female patients, implying that more regions contributed to stroke severity in women. These sex discrepancies were particularly pronounced for lesions affecting a number of left-hemispheric regions, i.e., thalamus, hippocampus, and occipital cortical brain regions, thus regions reminiscent of the vascular territory of the posterior circulation. Lesions in these regions were underpinning a substantially higher stroke severity in women only, indicating sex-specific distributions of functional deficits.

Our sex-specific findings are of interest from several points of view. First, we could not detect any sex differences in total normalized lesion volume, or in any normalized lesion volumes of the atlas-based brain regions and white matter tracts. Rather, similarly configured lesion patterns were linked to more severe strokes in women compared to men. Stroke severity in men was predominantly explained by five specific lesion patterns, denoting bilateral subcortical and left-sided insular and opercular regions. Women were characterized by similar, yet more pronounced relevant lesion patterns, as well as several additional relevant ones. These additional lesion patterns did not possess any relevance in men. Therefore, the female-specific effects can be considered more substantial and spatially widespread. Most noticeably, lesions in the presumed vascular territory of the left PCA, among other regions affecting left hippocampal and thalamic regions, left fusiform, lingual, and intracalcarine cortex, as well as left precuneus and cuneal cortex explained a disproportionally high stroke severity in women compared to men. This difference was consistently observable across datasets and did not appear to represent an artifact of a more frequent cardioembolic stroke genesis, more severe strokes or a higher age at stroke onset in women.

It is important to note that the strongest observed sex differences were strictly lateralized to the left hemisphere. Previous research suggests that male or female sex and respective sex hormones contribute to induce functional cerebral asymmetries[27]. Men appear to have a stronger hemispheric asymmetry; however, while robustly replicated, determined effect sizes have been small[28]. Such an enhanced asymmetry in men was also found in some early lesion studies on intelligence[29]. However, further early lesion studies suggested that lateralization differences between the sexes might be even more complex, i.e., female brains may be asymmetric to a comparable degree, yet in different ways[30,31]. In particular, it was found that left-hemispheric lesions in women led to both verbal and performance scale IQ deterioration, while only one quality—either verbal or performance—was affected in all other lesion and sex constellations[30,32]. Our outcome measure, NIHSS-based stroke

severity, cannot be broken down and thus does not allow for conclusions on specific functions, such as verbal and performance scale IQ. This level of granularity hampers a direct comparison to earlier studies. Nonetheless, we also find that particularly women are vulnerable to left-hemispheric lesions. Indeed, we can relate the most robust excess vulnerability of female vs. male patients to anatomically precise lesion locations in the left-hemispheric PCA territory, specifically featuring hippocampal, thalamic, and precuneal regions. Based on existing knowledge on these regions' physiological functions, it may be suggested that lesions in these regions more likely underlie (higher) cognitive, than, for example, basic motor functions. This predilection toward specific anatomical regions within the left hemisphere could also explain differences seen in earlier lesion studies, as these studies stratified patients only based on left- or right-sided lesion locations without investigating any greater spatial detail of lesions[29,30,32].

Altogether, a sex-specific lateralization effect, comparable to the considerable extent and spatial distribution of the one that we detected in two independent datasets, has not been described in previous lesion studies, nor in studies focused on sex discrepancies in functional cerebral asymmetries in healthy adults[28]. This discrepancy between previous reports and our findings may originate from (i) the greater anatomical resolution in our study compared to early sex-aware lesion studies, (ii) not explicitly considering sex in any of the recent lesion-symptom studies[18,21,33–35], and (iii) the fact that our lesion-symptom analysis jointly investigated effects of functional asymmetry and the capacity for acute compensation, e.g., by means of brain plasticity.

We furthermore observed signs of an interaction effect of sex with age, when stratifying the entire sample based on the median age at menopause[26]. None of the female-specific lesion pattern effects could be detected, when comparing men and women below the age of 52 years. In fact, we rather saw indications for a male-specific effect pertaining to left-hemispheric subcortical and insular brain regions in this age subgroup. In contrast, three lesion atoms, among others relating to the left-hemispheric PCA territory, were found to be more pronounced in men and women above the age of 52 years. This constellation was thus suggestive of a possible influence of sexual hormones, that are dramatically changed in women after menopause[36].

Sex differences in brain organization in general and in stroke incidence and outcome in particular are related to the influence of sex steroid hormones, with estrogen likely being the most prominent one[36,37]. These hormones are assumed to act via irreversible organizational and reversible activational mechanisms. The former effect implies the facilitation of definite male or female tissue phenotypes, while the latter requires the presence of the hormone for an effect[25]. Any sex difference that changes with menopause is thus rather linked to activational hormonal pathways. An example of sex differences in stroke thought to be due to activational hormonal effects is the reduced stroke risk due to the premenopausal cycle of estrogen. This protective effect is lost after menopause[38] and may not be reestablished by hormone replacement therapy independent of the time of initiation[39,40]. Experimental research has furthermore shown that female animals experience smaller stroke lesions than male animals[41]. This effect could be neutralized by ovariectomy and consequently a decrease in estrogen levels[42]. It was interpreted as hormone-linked sex-specific sensitivity to cerebral ischemia[25,29,43]. In fact, male-specific cell cultures of hippocampal neurons and astrocytes seem to be more vulnerable to ischemia than female-specific cell cultures[44], and even ischemia-independent research indicates an important role of estrogen in sustained hippocampal structural plasticity and associated cognitive function[45,46].

The effect of age that we witnessed here—stronger female-specific effects in older patients, yet none in younger patients—suggests an activational nature of the apparent sex disparities. While our outcome measure of global stroke severity does not allow for fine-grained evaluations of implicated functions, let alone hippocampus-linked cognitive functions, it is nonetheless notable that hippocampal regions, that were markedly susceptible to hormonal influences as outlined above, contributed to the female-specific lesion patterns identified in our study.

It is however important to note, that we did not have access to information on the hormonal status in women. Instead, we here assumed the same median age of menopause for all female patients. Employing the actual age at menopause, as well as increasing the number of younger patients in general would have allowed for stronger conclusions on the potential organizational or activational character of findings. Besides, the availability of the exact level of estrogens would be most desirable, as previous studies indicate that sex-specific functional cerebral asymmetries may even vary during the menstrual cycle[27]. Moreover, the measurement of exact hormonal levels could inform about the effects of hormonotherapy (e.g., hormone replacement therapy after menopause, oral contraceptives prior to menopause) and surgical interventions, such as hysterectomies, oophorectomies, and ovaries remaining after hysterectomy. Therefore, future studies could attempt to gather data on hormonal status, and also to recruit more balanced numbers of men and women of premenopausal and postmenopausal status. In addition, it might be promising to test links between implicated brain regions and sex-specific genetic underpinnings, as was recently introduced for healthy population samples[47,48].

Independent of their exact origin, the sex-sensitive effects that we observed in brain-behavior associations underlying stroke severity, could lead to important clinical implications. Since lesions of any kind explained more severe stroke in women, rescuing the same (normalized) amount of brain tissue—for example, by thrombolysis or mechanical thrombectomy—could have a more enhanced effect in female than male patients. This resulting expectation is well in line with previous reports on enhanced therapy response to intravenous thrombolysis[49–51] or more advantageous long-term outcome in women compared to men in clinical mechanical thrombectomy studies[52]. In this recently published study on sex discrepancies after thrombectomy[52], Sheth and colleagues ascertain these outcome differences between men and women despite comparable infarct volumes and reperfusion rates.

Most previous studies have relied on sex-independent, general cutoffs of salvageable tissue volumes to decide, for example, whether to undertake mechanical thrombectomy in the later time window[53]. A conceivable first step could be to revisit previous randomized clinical trials' data to investigate rescued tissue–sex interactions. Future clinical treatment studies could then prospectively and systematically test varying cutoffs for men and women, hypothesizing that rescuing a lesser amount of tissue in women would still be sufficient for a noticeable positive treatment response. Furthermore, it may be important to take into account this female-specific salvaging effect for lesions in the posterior circulation territory of the left hemisphere. Thrombectomy for primary distal posterior cerebral artery occlusion stroke was recently found to be a safe and potentially beneficial treatment option, as treated patients showed a pronounced decrease in stroke severity until discharge[54]. In view of these results, future thrombectomy studies could evaluate whether female patients benefited even more substantially from these reperfusion therapies of more distal PCA occlusions. Notably, such studies should also consider additional age–sex interactions, given that our ancillary analyses suggested more pronounced female-specific effects in case of more advanced age.

Overall, an effective step toward tailored medical stroke care[55,56] may lie in more sex-aware acute treatment decisions. Sex-specific guidelines on stroke prevention[7] could be complemented by sex-specific guidelines on acute treatment decisions, enhancing sex-sensitive stroke care and ultimately increasing the benefit for both men and women.

In this study, we explored sex disparities in lesion patterns of global stroke severity, which allows for some broad clinical implications of our findings. Naturally, our results may to a certain extent be dominated by effects due to motor symptoms, given their disproportionally high importance for the overall NIHSS score. To evaluate more specific brain functions at acute and chronic stages, future studies could therefore focus on sub-items of the NIHSS, such as language impairments, dysarthria, disturbances in orientation, neglect, or on specific cognitive behavioral tests, e.g., probing memory functions. This would be a promising approach to trace back our most prominent sex-specific finding, the relevance of the left PCA territory, to specific brain functions. Incorporating outcomes from more chronic time points would furthermore allow for more definite conclusions on long-term effects. Above all, it may be especially fruitful to examine whether sex differences observed here generalize to cerebral reorganization processes during the recovery phase post-stroke. In the positive case, generated results could fuel sex-specific personalized clinical rehabilitation endeavors[57].

Similarly, the spatial resolution of our Bayesian hierarchical approach stays within the realms of frequently arising, typical lesion patterns, and back-transformed atlas brain regions. Consequently, our region-wise approach does not allow for a comparably high spatial resolution of lesion-symptom analyses relying on voxel-wise data[18,34]. Nonetheless, this reduction in lesion dimensionality was necessary to render sex-aware analyses within our Bayesian hierarchical model framework feasible. We here employed two frequently used brain atlas templates[58,59] and unsupervised dimensionality reduction[22]. Future studies could explore differing atlas templates and also supervised strategies to organize brain regions. Such supervised approaches could reduce variability between lesion embeddings of different datasets, since we here saw subtle differences in automatically derived lesion atoms. Eventually, confidence in our results could then be increased in case of successful replication despite these methodological alterations. What is more, our Bayesian models could, for example, be modified to integrate an additional hierarchical level to differentiate between rich-club and non-rich-club regions[60,61] or contrast agglomerated and separately modeled sets of brain regions to test brain modes, as introduced by Godefroy and colleagues[62,63].

Lastly, we did carefully account for interindividual differences in important sociodemographic characteristics (e.g., age), comorbidities (e.g., atrial fibrillation), markers of chronic brain health (e.g., white matter hyperintensity lesion load), and stroke subtypes (e.g., cardioembolic stroke genesis). We furthermore adjusted all analyses for global, lesion atom-independent sex differences (e.g., due to a potentially longer time between symptom onset and hospital admission in women)[10]. In addition, each image and corresponding lesion mask was individually quality controlled to ensure the absence of overt spatial distortions (e.g., due to cerebral atrophies). However, our cohorts were slightly imbalanced with respect to the men:women ratio, which may not have faithfully captured the oftentimes reported equally high or higher stroke incidence in women[64,65]. Also, we did not have access to some measures, which could be of potential interest as covariate in our analysis, for a majority of patients; for example, delays in hospital arrival, the exact time between imaging and

NIHSS acquisition, acute changes in stroke severity (e.g., due to spontaneous reperfusion, brain edema, or seizures) or administered revascularization therapies and their interactions with stroke severity (e.g., NIHSS scores upon admission and after acute treatment may differ substantially). In this study, we aimed to address these factors indirectly by tailoring ancillary analyses to subgroups, e.g., only those patients whose data was acquired directly after admission. Their data acquisition thus preceded the onsets of any acute treatment effects with high probability—female-specific effects relating to left posterior circulation lesions were nonetheless reliably observable. Future large-scale studies are warranted to recruit equal numbers of men and women and systematically record further clinical aspects, such as delays in hospital arrival and acute treatments, to facilitate more comprehensive and explicit investigations. In particular, these studies may be enriched for patients receiving acute treatments to maximize the power of uncovering sex- and treatment-specific lesion atom effects. By these means, it may eventually be possible to quantify the clinical importance of sex-specific lesion patterns by taking into account their modifying effects on acute recanalization treatments. Since some evidence for example suggests sex-specific effects of white matter integrity on stroke outcome[66], further future studies may also not only include acute, but also chronic markers of brain health while taking sex into account more closely.

Women tend to have more severe strokes than men. Previous methodology did not allow to evaluate whether lesion patterns contributed to these sex differences in outcomes. By deriving a low-dimensional lesion representation and employing Bayesian hierarchical modeling, we here uncovered considerable sex discrepancies in lesion patterns explaining acute stroke severity. While bilateral subcortical and left-hemispheric inferior frontal, superior, and middle temporal regions, i.e., presumed motor and language regions, explained more severe strokes in both men and women, effects in women were more widespread and similar lesions underlay more severe strokes in women compared to men. In particular, lesions in the posterior circulation of the left hemisphere were associated with a higher stroke severity exclusively in women. These differences were robustly validated in a second independent, international multisite lesion dataset, and could not be explained by sex differences in lesion volume or more severe strokes in women in general.

## Methods

**Participant recruitment**. AIS patients ($n = 555$), considered as derivation cohort in this study, were admitted to Massachusetts General Hospital and enrolled as of part the Genes Associated with Stroke Risk and Outcomes Study (GASROS; mean age(SD): 65.0(14.8) years, 38% females, c.f., Table 1 for sex-specific numbers)[61,67]. Inclusion was generally considered for any AIS patient that met the following criteria; (i) adult patients ≥18 years of age, (ii) admitted to the emergency department with signs and symptoms of AIS, and (iii) neuroimaging confirmation of an acute infarct. Only patients with MRI data obtained within 48 h from admission, as well as complete phenotypic data, such as stroke severity and stroke risk factors were included in this study (i.e., complete case analyses, c.f., Supplementary Information for a sample size derivation). Patients gave written informed consent in accordance with the Declaration of Helsinki. The study protocol was approved by Massachusetts General Hospital's Institutional Review Board (Protocol #: 2001P001186).

**Stroke patient characteristics and imaging**. Patients were examined by trained, board-certified vascular neurologists. The recorded sociodemographic and clinical variables included age, sex, and common vascular risk factors (hypertension, diabetes mellitus type 2, atrial fibrillation, coronary artery disease). Stroke severity was captured by the acute NIHSS (0: no symptoms, 42: maximum stroke severity).

Each patient was scanned within 48 h of admission, standardized clinical imaging protocols included DWI (in the majority of cases on 1.5 T General Electric Signa scanners, and a few cases on 1.5 or 3 T Siemens scanners, repetition time (TR) 5000 ms, minimum echo time (TE) of 62–117 ms, field-of-view (FOV) 220 mm field-of-view, 5-mm slice thickness with a 1-mm gap, and 0 s/mm² (b-zero) and 1000 s/mm² b-values), and axial T2 FLAIR images (TR 5000 ms, minimum TE

of 62–116 ms, TI 2200 ms, FOV 220–240 mm). Ischemic DWI tissue lesions were manually outlined using semiautomated algorithms[68]. Raters were blinded to clinical outcomes.

**Magnetic resonance imaging: preprocessing**. Individual images were spatially normalized to standard Montreal Neurological Institute (MNI-152) space: we first linearly realigned both DWI and FLAIR images with an MNI template. Subsequently, we co-registered the DWI image to the FLAIR image, denoised the images[69], and lastly employed the unified segmentation algorithm to nonlinearly normalize the FLAIR image[70]. We masked lesioned tissue during this normalization step to mitigate the risk of image distortions[71]. The same transformation was applied to the DWI image, as well as the corresponding DWI-derived binary lesion mask. This preprocessing pipeline, especially featuring the co-registration of DWI and FLAIR images, was optimized to generate as reliable and accurate spatial normalizations for as many patients as possible. The quality of normalized DWI-lesion masks was carefully inspected by two experienced raters (A.K.B. and M.B). Insufficient quality, predominantly arising from moderate to severe motion artifacts and/or moderate to severe normalization errors led to the exclusion of patients (c.f. Supplementary Information for details). The WMH volume (WMHv) was computed based on manual WMH outlines onto FLAIR images performed using the MRIcro software (04/2010, 08/2014, and 2019 versions)[72].

**Derivation of a low-dimensional lesion representation**. We successively combined (1) an automated low-dimensional embedding of high-dimensional lesion information, and (2) probabilistic modeling to explain the acute stroke severity, as measured by the NIHSS (c.f. ref. [23], for a comparable analytical approach). While we initially determined eloquent lesion patterns across all patients, we subsequently refined analyses to investigate sex differences on different levels of our Bayesian hierarchical model.

In view of the high-dimensional lesioned voxel space that would inevitably lead to severe overfitting in our regression analyses, we first captured the number of lesioned voxels within 109 brain regions, as defined by the Harvard-Oxford atlas[58]. More precisely, these brain regions represented 47 cortical and 7 subcortical gray matter brain regions per hemisphere, as well as the brainstem. White matter damage was read out in form of lesion load within each one of 20 John-Hopkins-University (JHU) atlas defined white matter tracts (7 hemisphere-specific tracts, 6 bilateral tracts)[59,73]. Thus, these steps resulted in 129 parcels in total. To extract reoccurring, archetypical and directly interpretable lesion patterns in our stroke population in a data-driven, unsupervised fashion, we performed NMF[22] on the log-transformed brain region- and tract-wise lesion load. By these means, we obtained ten unique topographical lesion configurations, that we will call lesion atoms in the following (c.f., Supplementary Information for further details). Probable vascular territories of lesion atoms were assigned via evaluations by individuals with several years of neurology and neuroradiology experience (A.K.B., M.B., and N.R.).

**Explaining interindividual differences in acute stroke outcomes**. These ten NMF-derived lesion atoms, containing information on individual lesion patterns, served as neuroimaging-derived input to our Bayesian hierarchical linear regression model[74] to explain acute stroke severity. We aimed to obtain fully probabilistic model parameter estimates that could inform us about each lesion atom's influence on the outcome. While we first computed the impact of lesion atoms across all patients in a first Bayesian model, we refined analyses in a subsequent step and introduced a hierarchical lesion atom structure for a second model. This hierarchy allowed the stratification for an individual's sex, i.e., we could estimate the lesion atom's influence on the outcome in women and men separately. Hence, we obtained separate lesion pattern relevances for men and women.

Prior to carrying out the Bayesian model, lesion atom data, as well as the stroke severity outcome score were corrected for lesion volume[75]. In addition, our model took into account the effects of (normalized) age, age², sex, and the presence of the following comorbidities: hypertension, diabetes mellitus type 2, atrial fibrillation, coronary artery disease, and lastly the log-transformed white matter hyperintensity lesion volume. In view of women's overall higher age, we aimed to adjust for age-specific effects particularly comprehensively by not only taking into account the basic age, yet also its squared value and therefore U-shaped effects as well. Importantly, we also included sex as a nonhierarchical, lower-level variable in the model to capture sex differences in stroke severity that were independent of lesion patterns. Women may, for example, also present with more severe strokes on average. They may be of a more advanced age and have a greater prestroke level of disability when experiencing the stroke, leading to a higher stroke severity, yet decreased likelihood of acute recanalization treatment. Such sex-specific, but lesion pattern-independent differences in stroke severity could then be modeled by the sex variable. It would, however, not have an influence on lesion pattern–sex interaction effects. In other words, the nonhierarchical sex variable represents the "residual" sex effects on the outcome that are not already captured by the hierarchical sex-specific NMF components.

The full Bayesian hierarchical model specification was as follows:
**Hyperpriors**
$hyper\_\sigma\_\ \beta \sim Halfcauchy(5)$

$\sigma\_\beta_{m,f} \sim Halfcauchy(hyper\_\sigma\_\beta)$

$hyper\_mu\_\beta \sim Normal(\mu = 0, \sigma = 10)$

$mu\_\beta_{m,f} \sim Normal(\mu = hyper\_mu\_\beta, \sigma = 10)$

**Priors**

$\alpha \sim Normal(\mu = 0, \sigma = 1)$

$\beta_{1-10;\ m,f} \sim Normal(\mu = mu\_\beta_{m,f}, \sigma = \sigma\_\beta_{m,f})$

$\beta_{age} \sim Normal(\mu = 0, \sigma = 10)$

$\beta_{age*age} \sim Normal(\mu = 0, \sigma = 10)$

$\beta_{sex} \sim Normal(\mu = 0, \sigma = 1)$

$\beta_{hypertension} \sim Normal(\mu = 0, \sigma = 1)$

$\beta_{diabetes} \sim Normal(\mu = 0, \sigma = 1)$

$\beta_{atrial\ fibrillation} \sim Normal(\mu = 0, \sigma = 1)$

$\beta_{coronary\ artery\ disease} \sim Normal(\mu = 0, \sigma = 1)$

$\beta_{WMHv} \sim Normal(\mu = 0, \sigma = 1)$

**Likelihood of linear model**

$NIHSS\_est = \alpha + \beta_{1-10}[sex]* NMF\text{-}Component_{1-10} + \beta_{age}* Age + \beta_{age*age}* Age^2 + \beta_{sex}* Sex + \beta_{hypertension}* hypertension + \beta_{diabetes}* diabetes + \beta_{atrial\ fibrillation}*$ atrial fibrillation $+ \beta_{coronary\ artery\ disease}*$ coronary artery disease $+ \beta_{WMHv}*$ WMHv

**Model likelihood**

$eps \sim Halfcauchy(20)$

$stroke\_severity \sim Normal(\mu = NIHSS\_est, \sigma = eps)$

We employed the No U-Turn Sampler (NUTS), a type of Monte Carlo Markov Chain algorithm (setting: draws = 5000)[76], to draw samples from the Bayesian posterior parameter distributions.

Main advantages of our analytical approach may be seen in (i) the enhanced interpretability of estimated NMF components, i.e., lesion atoms, (ii) the multivariate nature of the stroke lesion embedding, and (iii) the possibility of integrating interaction effects between lesion atoms and an individual's sex in a statistically sound probabilistic modeling framework. We thus make a step towards providing an alternative approach to classical VLSM analyses[18], that estimate the magnitude of effects in one brain location at a time and thus fit one model per voxel. Our approach opens an alternative window to "one voxel at a time" by fitting a single model to the entire brain. All ten of our lesion atoms were entered into the regression model at once. Hence, we could estimate the effect of each lesion atom, while accounting for the effects of all further lesion atoms. By these means, our approach resembles more machine-learning-based multivariate lesion-symptom mapping methods, that have been introduced recently to evaluate lesion patterns instead of single voxels in association to an outcome[77]. Examples can be seen in approaches leveraging support vector machines[34,78,79], tree-based[80], or game theory-based algorithms[81]. Importantly, all of these multivariate approaches may decrease the distortion of functional localization compared to univariate voxel-wise approaches by considering subcortically and cortically located lesions at once[20,82]. We additionally aimed to mitigate confounding effects of excessively correlated neighboring voxels, as naturally arising due to the stroke lesion-vasculature dependence, by explicitly combining frequently jointly affected brain regions in lesion patterns, i.e., our lesion atoms. Despite these methodological innovations, no currently existing stroke modeling approach may achieve unfailing causal conclusions[20].

**Validation analysis**. We repeated analyses in an independent dataset of 503 ischemic stroke patients (mean age(SD): 65.0(14.6), sex: 40.6% female, mean NIHSS(SD): 5.48(5.35), c.f., Supplementary Table 1 for further clinical characteristics), acquired within the framework of the multisite MRI-GENIE study[24], to test the robustness of our findings (c.f., Supplementary Information for details). To further ascertain the robustness of our unsupervised lesion embedding, we correlated NMF components as computed for the derivation and validation cohort. Moreover, we recomputed region-wise relevances in final nonhierarchical analyses based on the other cohort's lesion embedding (i.e., we first estimated the NMF transformation based on one cohort's imaging data and then applied this transformation to the data of the other cohort). We then ran additional correlation analyses between region-wise relevances as originating from the cohort-specific embedding, as well as the other cohort's embedding to ensure our results were not substantially altered by the particular embedding.

**Ancillary analyses**. We performed three ancillary hierarchical analyses. Firstly, we aimed to reduce potential confounds arising from dispersed time points of data acquisition and effects of acute recanalization therapies and limited the derivation cohort sample to only those patients whose data was acquired directly upon admission (and not within the first 48 h, as for the entire cohort). Thus, their data acquisition likely preceded the potential administration of any acute recanalization therapy and their lesion-symptom associations may be considered to be purer.

Conceivably, female-specific lesion effects could be due to a higher frequency of cardioembolic strokes and associated typical lesion patterns (e.g., more multifocal stroke lesions)[83]. In a second ancillary analysis, we therefore sought to disentangle stroke-subtype- and sex-specific effects, and concurrently performed stroke-subtype- and sex-aware analyses, featuring two hierarchical levels. Accordingly, patients were not only stratified into groups of men and women as in main analyses, yet initially assigned to a group of either cardioembolic or non-cardioembolic stroke genesis, and then split into men and women (c.f., Supplementary Information for a comprehensive display of model specifications).

We then contrasted male- and female-specific lesion pattern effects within the cardioembolic and non-cardioembolic stroke subgroups. To allow for a reasonably high number of patients in each of these four subgroups, we here merged data of the derivation and validation cohorts (c.f., Supplementary Information for further details).

Lastly, we aimed to explore whether possible sex differences were more likely to be of organizational or activational hormonal nature. Since we did not have access to the precise hormonal status in women, we stratified our sample based on an age cutoff of 52 years, according to the median age at menopause[26]. As in the ancillary analysis focused on stroke subtype, we inserted an additional hierarchical level and build two groups of patients above ($\geq$) 52 and below (<52) years of age within which we compared lesion pattern effects of men and women. We once again merged derivation and validation cohort data to maximize the number of available patients, especially in the subgroup of younger stroke patients.

**Reporting summary**. Further information on research design is available in the Nature Research Reporting Summary linked to this article.

## Data availability

The authors agree to make the data of development and validation cohorts available to any researcher for the express purposes of reproducing the here presented results, and with the explicit permission for data sharing by Massachusetts General Hospital's institutional review board. The Harvard-Oxford and JHU DTI-based white matter atlases are accessible online (https://fsl.fmrib.ox.ac.uk/fsl/fslwiki/Atlases). Source data are provided with this paper.

## Code availability

Preprocessing of MRI images was conducted in a Matlab 2019b framework (The Mathworks, Natick, MA, USA), the packages Statistical Parametric Mapping (SPM12; http://www.fil.ion.ucl.ac.uk/spm/), and the ancillary package SPM_Superres (https://github.com/brudfors/spm_superres). Further analyses were implemented in Python 3.7 (primarily packages: nilearn[84] and pymc3; ref. [85]). Full code for reproducibility and reuse is available here: https://github.com/AnnaBonkhoff/BMH_stroke_severity_sex_differences[86].

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

## Acknowledgements

We are grateful to our colleagues at the J. Philip Kistler Stroke Research Center for valuable support and discussions and would like to specifically acknowledge valuable comments by Parashkev Nachev, M.D., Ph.D. on previous versions on this manuscript. Furthermore, we are grateful to our research participants without whom this work would not have been possible. M.B. acknowledges support from the Société Française de Neuroradiologie, Société Française de Radiologie, and Fondation ISITE-ULNE. A.V. is in part supported by NIH-NINDS (R01 NS103824, RF1 NS117643, R01 NS100417, U01NS100699, and U01NS110772). C.J. acknowledges support from the Swedish Research Council (2018-02543), the Swedish state under the agreement between the Swedish government and the county councils, the ALF agreement (ALFGBG-720081); the Swedish Heart and Lung Foundation (20190203). A.G.L. acknowledges support from the Swedish Research Council (2019-01757), The Swedish Government (under the "Avtal om Läkarutbildning och Medicinsk Forskning, ALF"), The Swedish Heart and Lung Foundation, Region Skåne, Lund University, Skåne University Hospital, Sparbanks-stiftelsen Färs och Frosta, Fremasons Lodge of Instruction Eos in Lund and NIH (1R01NS114045-01). D.B. has been funded by the Brain Canada Foundation, through the Canada Brain Research Fund, with the financial support of Health Canada, National Institutes of Health (NIH R01 AG068563A), the Canadian Institute of Health Research (CIHR 438531), the Healthy Brains Healthy Lives initiative (Canada First Research Excellence fund), Google (Research Award, Teaching Award), and by the CIFAR Artificial Intelligence Chairs program (Canada Institute for Advanced Research). N.S.R. is in part supported by NIH-NINDS (R01NS082285, R01NS086905, and U19NS115388).

## Author contributions

A.K.B., D.B., and N.S.R. conceived and designed the study, led data interpretation, and prepared the manuscript. A.K.B. led data analysis, M.D.S., M.B., S.H., R.W.R., M.B., and O.W. also contributed to data analysis. E.D. led data management, data preprocessing, and execution of data analysis workflows. K.L.D., M.J.N., A.V.D., A.K.G., M.R.E., B.L.H., B.J.T.M., E.C.M., J.A., O.S.B., S.B., J.W.C., A.D., C.J.G., La.H., Lu.H., K.J., J.J.C., S.J.K., R.L., C.R.L., C.W.M., J.F.M., C.L.P., A.R., S.R., Jo.R., Ja.R., T.R., R.L.S., R.S., P.S., Ag.S., M.S., Al.S., T.M.S, D.S., T.T., V.T., A.V., J.W., D.W., R.Z., P.F.M., B.B.W., C.J., A.G.L., J.M., and O.W. contributed to data acquisition, management, and preprocessing. All authors contributed to results interpretation and final manuscript preparation.

## Competing interests

M.E. has received personal fees for consulting from Astra Zeneca and WorldCare Clinical Group. C.G. has received consulting honoraria from Microvention and Strykere, and research funding from Medtronic and Penumbra. A.V. has received research funding from Cerenovus. A.G.L. has received personal fees from Bayer, Astra Zeneca, BMS Pfizer, and Portola. N.S.R. has received compensation as scientific advisory consultant from Omniox, Sanofi Genzyme, and AbbVie Inc. All other authors declare no competing interests.

## Additional information

[1]J. Philip Kistler Stroke Research Center, Massachusetts General Hospital, Harvard Medical School, Boston, MA, USA. [2]Clinic for Neuroradiology, University Hospital Bonn, Bonn, Germany. [3]Univ. Lille, Inserm, CHU Lille, U1171 – LilNCog (JPARC) – Lille Neurosciences & Cognition, F-59000, Lille, France. [4]School of Biomedical Engineering & Imaging Sciences, King's College London, London, UK. [5]Computer Science and Artificial Intelligence Lab, Massachusetts Institute of Technology, Boston, MA, USA. [6]Athinoula A. Martinos Center for Biomedical Imaging, Department of Radiology, Massachusetts General Hospital, Charlestown, MA, USA. [7]Department of Neurology, University Medical Center Hamburg-Eppendorf, Hamburg, Germany. [8]Hunter Medical Research Institute, Newcastle, NSW, Australia. [9]School of Medicine and Public Health, University of Newcastle, Newcastle, NSW, Australia. [10]Department of Medicine, Division of Neurology, University of British Columbia, Vancouver, BC, Canada. [11]School of Life Sciences, University of Lincoln, Lincoln, UK. [12]Department of Neurology, University of Maryland School of Medicine and Veterans Affairs Maryland Health Care System, Baltimore, MD, USA. [13]School of Medical Sciences, University of Campinas (UNICAMP) and the Brazilian Institute of Neuroscience and Neurotechnology (BRAINN), Campinas, Sao Paulo, Brazil. [14]Department of Neurosurgery, Geisinger, Danville, PA, USA. [15]Research Institute of Neurointervention, Paracelsus Medical University, Salzburg, Austria. [16]Department of Emergency Medicine, Washington University School of Medicine, St Louis, MO, USA. [17]Department of Neurology, Washington University School of Medicine & Barnes-Jewish Hospital, St Louis, MO, USA. [18]Department of Clinical Neuroscience, Institute of Neuroscience and Physiology, Sahlgrenska Academy, University of Gothenburg, Gothenburg, Sweden. [19]Department of Neurology, Sahlgrenska University Hospital, Gothenburg, Sweden. [20]Department of Neurology, Neurovascular Research Group (NEUVAS), IMIM-Hospital del Mar (Institut Hospital del Mar d'Investigacions Mèdiques), Universitat Autonoma de Barcelona, Barcelona, Spain. [21]KU Leuven - University of Leuven, Department of Neurosciences, Experimental Neurology and Leuven Research Institute for Neuroscience and Disease (LIND), Leuven, Belgium. [22]VIB, Vesalius Research Center, Laboratory of Neurobiology, University Hospitals Leuven, Department of Neurology, Leuven, Belgium. [23]School of Medicine and Public Health, University of

Newcastle, Newcastle, NSW, Australia. [24]Department of Neurology, John Hunter Hospital, Newcastle, NSW, Australia. [25]Department of Pharmacotherapy and Translational Research and Center for Pharmacogenomics, University of Florida, Gainesville, FL, USA. [26]Department of Neurology, Mayo Clinic, Jacksonville, FL, USA. [27]Centogene AG, Rostock, Germany. [28]Department of Neurology, Clinical Division of Neurogeriatrics, Medical University Graz, Graz, Austria. [29]Center for Genomic Medicine, Massachusetts General Hospital, Boston, MA, USA. [30]Department of Neurology and Evelyn F. McKnight Brain Institute, Miller School of Medicine, University of Miami, Miami, FL, USA. [31]Institute of Cardiovascular Research, Royal Holloway University of London (ICR2UL), Egham, UK. [32]St Peter's and Ashford Hospitals, Egham, UK. [33]Department of Neurology, Jagiellonian University Medical College, Krakow, Poland. [34]Department of clinical sciences Malmö, Lund University, Lund, Sweden. [35]Department of Neurology, Skåne University Hospital, Lund and Malmö, Sweden. [36]Department of Laboratory Medicine, Institute of Biomedicine, the Sahlgrenska Academy at University of Gothenburg, Gothenburg, Sweden. [37]Department of Neurology, Helsinki University Hospital and University of Helsinki, Helsinki, Finland. [38]Department of Clinical Neuroscience, Institute of Neuroscience and Physiology, Sahlgrenska Academy at University of Gothenburg, Gothenburg, Sweden. [39]Department of Neurology, Sahlgrenska University Hospital, Gothenburg, Sweden. [40]Stroke Division, Florey Institute of Neuroscience and Mental Health, Heidelberg, Australia. [41]Department of Neurology, Austin Health, Heidelberg, Australia. [42]Department of Radiology, University of Cincinnati College of Medicine, Cincinnati, OH, USA. [43]Department of Clinical Sciences Lund, Radiology, Lund University, Lund, Sweden. [44]Department of Radiology, Neuroradiology, Skåne University Hospital, Lund, Sweden. [45]Department of Neurology and Rehabilitation Medicine, University of Cincinnati College of Medicine, Cincinnati, OH, USA. [46]Department of Neurology, Geisinger, Danville, PA, USA. [47]Division of Endocrinology, Diabetes and Nutrition, Department of Medicine, University of Maryland School of Medicine, Baltimore, MD, USA. [48]Departments of Neurology and Public Health Sciences, University of Virginia, Charlottesville, VA, USA. [49]Department of Clinical Genetics and Genomics, Sahlgrenska University Hospital, Gothenburg, Sweden. [50]Department of Neurology, Skåne University Hospital, Lund, Sweden. [51]Department of Clinical Sciences Lund, Neurology, Lund University, Lund, Sweden. [52]University of Technology Sydney, Sydney, NSW, Australia. [53]Department of Biomedical Engineering, McConnell Brain Imaging Centre, Montreal Neurological Institute, Faculty of Medicine, School of Computer Science, McGill University, Montreal, QC, Canada. [54]Mila - Quebec Artificial Intelligence Institute, Montreal, QC, Canada. *A list of authors and their affiliations appears at the end of the paper. ✉email: abonkhoff@mgh.harvard.edu

## MRI-GENIE and GISCOME Investigators and the International Stroke Genetics Consortium

Anna K. Bonkhoff[1✉], Markus D. Schirmer[1,2], Martin Bretzner[1,3], Sungmin Hong[1], Robert W. Regenhardt[1], Mikael Brudfors[4], Kathleen L. Donahue[1], Marco J. Nardin[1], Adrian V. Dalca[5,6], Anne-Katrin Giese[7], Mark R. Etherton[1], Brandon L. Hancock[6], Steven J. T. Mocking[6], Elissa C. McIntosh[6], John Attia[8,9], Oscar R. Benavente[10], Stephen Bevan[11], John W. Cole[12], Amanda Donatti[13], Christoph J. Griessenauer[14,15], Laura Heitsch[16,17], Lukas Holmegaard[18,19], Katarina Jood[18,19], Jordi Jimenez-Conde[20], Steven J. Kittner[12], Robin Lemmens[21,22], Christopher R. Levi[23,24], Caitrin W. McDonough[25], James F. Meschia[26], Chia-Ling Phuah[17], Arndt Rolfs[27], Stefan Ropele[28], Jonathan Rosand[1,6,29], Jaume Roquer[20], Tatjana Rundek[30], Ralph L. Sacco[30], Reinhold Schmidt[28], Pankaj Sharma[31,32], Agnieszka Slowik[33], Martin Söderholm[34,35], Alessandro Sousa[13], Tara M. Stanne[36], Daniel Strbian[37], Turgut Tatlisumak[38,39], Vincent Thijs[40,41], Achala Vagal[42], Johan Wasselius[43,44], Daniel Woo[45], Ramin Zand[46], Patrick F. McArdle[47], Bradford B. Worrall[48], Christina Jern[36,49], Arne G. Lindgren[50,51], Jane Maguire[52], Danilo Bzdok[53,54], Ona Wu[6] & Natalia S. Rost[1]

