## [Peer Review File · Nature Communications]

REVIEWER COMMENTS

Reviewer #1 (Remarks to the Author):

The present study reports a combined lesion symptom mapping (LSM) and Bayesian analysis conducted in 555 patients (and replicated in other 503 patients) in order to explore male and female-specific lesion patterns underlying stroke severity as assessed with NIHSS. This analysis was able (1) to uncover a more widespread pattern of relevant lesions in women than in men, (2) to emphasize the importance of lesions in the left hemisphere PCA territory in generating higher stroke severity in women and (3) to suggest sex-specific functional cerebral asymmetries to be further used in acute stroke therapies. This study focuses on an interesting topic seldom investigated explicitly and therefore could be of significance to the field. Unfortunately, several points are difficult to follow (also because of the novelty of the method) and should be more clearly or explicitly stated:

1) The method seems interesting, but not enough details are provided and the reader is referred to another paper not peer reviewed yet... For instance, it is not clear how authors dealt with well-known LSM problems such as the impact of vascular territories in lesions localization, or data correlations among neighboring voxels (see also Mah et al., 2014; Sperber, 2020; Toba et al., 2020; Vaidya et al., 2019), to mention just a few. Authors specify that lesion atoms 6 and 10 denoted the vascular supply territory of the posterior circulation. This can be an artifact! Please explain how the role of the vascular territories was controlled. It is not clear either whether authors considered variables such as cerebral atrophy known to play a role in post-stroke disorders (see Rothwell et al., 2005) such as those assessed with NIHSS. Also, how this method is situated compared with multivariate methods that begin to be used in the field (e.g., Smith et al., 2013; Zhang et al., 2014; Yourganov et al., 2016; Toba et al., 2017; Pustina et al., 2017)? All these details should be mentioned in the manuscript so that the reader can understand the method without referring to other preprints.

2) Authors should better emphasize that analyses were conducted in two separated (derivation and replication) groups and that results were similar, although some differences could be observed. For instance, how authors explain that in the replication analyses, 5 lesion atoms have been found in the right hemisphere, while only 4 have been reported for the derivation group? This is an important issue because it shows the instability of behavioural and lesional patterns even when analyses are conducted in more than 500 patients. This information should also be present in the abstract.

3) Authors used both Harvard Oxford and JHU atlases, but they do not explain how they dealt with overlapping voxels. This is an issue with important consequences on statistical analyses. Why did the authors not use a template integrating both grey and white matter?

4) For the second (replication) group, only scarce information is provided (age: 65.0 (14.6), sex: 40.6% female, NIHSS: 5.48 (5.35)). The NIHSS score provided here represents the mean, or the median (as provided for the derivation group in Table 1)? Please include for the replication group a table with the same information as for the derivation group, so that the reader can compare data.

5) An important part of the NIHSS score is represented by motricity. Authors should better emphasize this point and precise that current lesion atoms are biased by the use of such a scarce scale to evaluate behaviour.

6) It is not clear why authors did correct for the sex covariate in the first analyses (line 219), and considered it in the model only in a second step.

7) In the first group, age was significantly different between men and women. Can this element have an impact on potential cerebral atrophies for instance? How this issue has been controlled in the analysis?

8) Why is an uncorrected p provided in supplementary tables 1 and 2, while in the text it is mentioned that analyses are Bonferroni corrected (line 196)? Please be coherent.

9) Discussion could be further developed to integrate the translational neuroscience aspect of these findings and particularly the personalized clinical rehabilitation.

Minor:

Please be consistent with terms used for the first group. Currently both 'derivation' and 'development' terms are used.

The title of the fig 3 can be misleading 'Sex-specific posterior distributions'. Please consider to define the term 'posterior' here, that could be interpreted as related to posterior anatomical territories.

Please correct the reference 23 in 2,3 (line 131) and 'globus pallidum' (line 230)

Reviewer #2 (Remarks to the Author):

Sex-specific lesion topographies explain outcomes after acute ischemic stroke

Anna K. Bonkhoff et al. 2020

Peer Review for Nature Communications,

This a compelling article on gender-specific functional cerebral asymmetries and gender-stratified deficit after acute ischemic stroke. The authors test gender differences in lesion patterns that explain stroke severity in two large independent datasets. They find that females show a more widespread pattern of lesions explaining acute severity. They argue that lesions in the left hemisphere's posterior circulation relate to a higher stroke severity exclusively in women. According to the authors, the findings suggest gender-specific functional cerebral asymmetries, which would motivate a more gender-sensitive treatment decision approach.

The study addresses a potentially high impact issue in the comprehension of gender differences in the prediction of stroke severity that may contribute to the development of gender-specific stroke therapies. The manuscript is well written, the cohort is adequate, and the methodology is potentially proper, but several significant caveats require attention. Some of these issues may not be solvable, which could impact the main message.

Major critiques

1. The stroke dataset may not be representative of a clinically relevant sample. The NIHSS scores are overall low (mean 3), which is not typical of a prospective sample where the mean/median is around 7. The Authors should discuss inclusion/exclusion criteria and the presence of possible biases of selection. The authors do not specify the exact timing of NIHSS score recordings vis-a-vis the onset of stroke. "Within 48h from admission" is not acceptable, primarily as they do not provide any information on acute revascularization treatment of these patients (e.g., is the NIHSS before or after treatment?). The NIHSS scores can enormously change in the first hours from stroke onset depending on various factors, including micro-embolic signals, spontaneous reperfusion, blood pressure and perfusion pressure, metabolic homeostasis, brain edema, or seizure, etc., and these changes differ according to etiology (see next point).
2. Women are known to go later to the ER, which could bias the results: was this data collected?
3. The etiology of stroke is missing. The prevalence of different etiologies is different between men and women (i.e., cardioembolic vs. other causes), which would affect the anatomical distribution of lesions.
4. It is not clear whether stroke lesions were focal or multifocal in each subgroup. This information typically relates to differences in stroke etiology --multifocal lesions are embolic--and could also impact stroke severity compared to focal lesions.
5. Were the NIHSS and MRI data collected in close temporal proximity? If not, this could also affect the results in the lesion-behavior correlation analysis.
6. The authors state DWI and FLAIR were co-registered for segmentation. Did these scans follow treatment? Were they performed in the hyper-acute setting for diagnosis? An MRI scan performed 2 hours after stroke onset cannot rely on a FLAIR sequence differently from DWI. The two sequences are more alligned at 48 hrs.
7. The authors compare male vs. female patients, and found no statistical differences in lesion

volume. However, it is not clear whether the frequency maps of lesions for male and females is similar. In other words, while the number of voxels affected in each parcel is not statistically different, do female and male lesions affect the same areas with the same frequency? This data and frequency maps for each subgroup need inclusion in the analysis (ideally displayed from 0 to maximum value).

8. This consideration is equally valid when the authors explore the effects of gender hormones on gender-specific differences, using a demographic categorization (age cutoff of 52 years). The larger effect in the older sample could be due to other factors such as a higher incidence of AF in older women. It is known that cardioembolic strokes affect more the ACA and PCA distribution (lesion atom 1 and 10).

9. Since the main difference is in the left posterior (parietal) cortex, it is necessary to analyze the sub-scores of the NIHSS more related to the function carried out in that region. Language, dysarthria, and orientation scores should be more predictive than motor scores of the difference in lesion distribution.

10. The NIHSS scores' distribution should be displayed (in supplementary) separately for male, female, and female older and younger than 52.

Overall, many substantial issues affect the main conclusion that differences in lesion distribution for males and females depend on brain substrates. This conclusion is by exclusion after all other factors mentioned above rule out. Since negative findings carry not as much weight as positive findings, it is necessary to rule out all issues above systematically before a conclusion about different brain substrates is acceptable with some confidence.

Minor:

1. The two datasets are unbalanced in terms of the ratio of male/female (more females). Why? This difference is surprising in a random sample.

2. Was the stroke severity statistically higher in the female population based on the NIHSS scores? In both datasets? Please analyze.

3. The population size in the abstract is misleading. The authors should specify that the sample size was ~500, and additional 500 subjects were part of a validation cohort.

4. A region-wise approach rather than a voxel-wise approach reduces the anatomical selectivity of the results. This issue needs discussion in the "Limitations and future directions" paragraph.

Overall a provocative paper, but the main conclusions and most exciting conclusions are not robust yet.

Reviewer #3 (Remarks to the Author):

This study provides interesting data on the spatial distributions of stroke in men versus women, and how these differences contribute to stroke severity measured by the NIHSS. There are many strengths, including novel methodology (using Bayesian hierarchical linear regression), a large database, replication in a separate database. However, there are also some important weaknesses.

First, many of the references are outdated, rendering some of the background "facts" incorrect. For example, the authors state, "For instance, women have a lower stroke incidence than men when younger, yet this initially low women-to-men incidence ratio is decisively reversed in the oldest age groups (>85 years).

However, according the AHA website, women have a higher incidence of stroke than me.

<https://www.stroke.org/en/about-stroke/stroke-risk-factors/women-have-a-higher-risk-of-stroke>
See also:

Madsen, T.E., Khoury, J., Alwell, K., Moomaw, C.J., Rademacher, E., Flaherty, M.L., Woo, D., Mackey, J., La Rosa, F.D.L.R., Martini, S. and Ferioli, S., 2017. Sex-specific stroke incidence over time in the Greater Cincinnati/Northern Kentucky Stroke Study. *Neurology*, 89(10), pp.990-996. These authors report, "Contrary to previous study periods, stroke incidence rates were similar by sex in 2010."

See also:

Giroud, M., Delpont, B., Daubail, B., Blanc, C., Durier, J., Giroud, M. and Béjot, Y., 2017. Temporal

trends in sex differences with regard to stroke incidence: the Dijon stroke registry (1987–2012). *Stroke*, 48(4), pp.846-849.

These authors conclude, "...more women than men experience an incident stroke each year because of a longer life expectancy."

Furthermore, the age-adjusted incidence in women is also higher than men in some countries:

Kim, J., Thayabaranathan, T., Donnan, G.A., Howard, G., Howard, V.J., Rothwell, P.M., Feigin, V., Norrving, B., Owolabi, M., Pandian, J. and Liu, L., 2020. Global stroke statistics 2019. *International Journal of Stroke*, p.1747493020909545.

These authors report, "The sex-specific age-adjusted incidence in China was greater in women than in men (309 vs. 280 per 100,000 population; online Supplementary Table 6)."

The higher incidence of stroke in women in most countries, together with the inclusion of only 37% women in the population studied raises the possibility of an important bias. That is, women with less symptomatic strokes may not have been included (because they were not identified as having stroke).

More importantly, the findings that both posterior (inferior division) MCA stroke and left PCA stroke were more common in women likely reflects the greater rate of atrial fibrillation in women. Atrial fibrillation is known to cause larger strokes as well as more frequent inferior division (compared to superior division) MCA strokes. This result accounts for the greater frequency of Wernicke's aphasia than Broca's aphasia in cardioembolic stroke. This result has been attributed to the greater accessibility of the inferior division than superior division to clots from the heart. Likewise, atrial fibrillation results in higher rate of posterior circulation (e.g. PCA) strokes than other common etiologies (such as carotid disease). Carotid stenosis (more common in men than women) is more likely to cause anterior (superior division) strokes.

Therefore, the results need to be re-analyzed by stroke etiology to determine if the findings can really be attributed to sex differences. The role of hormones is interesting, but purely speculative. The finding of stronger sex-specific effects in older patients and no sex-specific effects in younger patients might again reflect differences in etiology of stroke (more atrial fibrillation in women) rather than sex differences.

Reviewer #4 (Remarks to the Author):

The authors have performed detailed analyses of sex-specific lesions in men and women with acute ischemic stroke. They claim a novel interpretation for why women may have more severe strokes (based on NIHSS) compared to men, and this relates to specific regions of the brain that differ by sex. The authors also performed analyses by stratifying by age, choosing 52 yrs as the strata due to the age at menopause.

Overall, these results are very fascinating since this detailed mapping of lesions based on severity scores clearly demonstrate sex differences. I have concerns that relate to the hormonal status analyses.

1. Some women use hormone therapy past age 52, so the limitations need to include this fact. Also, many women under age 52 could have been using oral contraceptives. There are also nuances related to age at hysterectomy/oophorectomy and ovaries remaining after hysterectomy that need to be acknowledged.
2. In addition, the women were older than men at the time of the stroke, and it is not clear how this was taken into account in the overall analyses.
3. The low prevalence of hypertension was quite surprising--only 28%?
4. Understanding the impact of this imaging analysis would be quite powerful if there were functional clinical outcomes at 30 or 90 days. Perhaps this is the follow-up plan in these cohorts, but it would be a nice validation of cognitive, fatigue, or other quality of life outcomes that could be compared between men and women.

Dear Referees,

We thank you for careful consideration of our manuscript, “Sex-specific lesion topographies explain outcomes after acute ischemic stroke”.

We believe that the thoughtful suggestions have enabled us to substantially improve the manuscript. In particular, we now disentangle sex-specific effects from stroke-subtype-specific effects. Within the framework of newly added ancillary analyses, we ensure that the observed sex differences do not arise due to varying frequencies of cardioembolic strokes between men and women.

We would like to acknowledge at this point that we realized a minor coding error upon investigating the origin of varying numbers of lesion atoms per hemisphere, as highlighted by referee 1, question 2. For the derivation cohort, we had inadvertently integrated lesion data for cortical atlas regions that was normalized to the same brain region size, and not to each individual brain region size, as was the case for subcortical and white matter tract regions.

Results remained very similar after incorporating the harmonized lesion data. However, the NMF-dimensionality-reduction-step was now stabilized, and we observed an equal number of lesion atoms per hemisphere (i.e., five per hemisphere and not four in the left and six in the right hemisphere, as previously). Furthermore, the more widespread and more pronounced lesion patterns in women were rendered even more apparent: Substantial effects were now evident for five, instead of two lesion atoms.

Importantly, the lesion pattern relating to left-hemispheric thalamic, hippocampal and occipital brain regions continued to be the most pronounced and robustly emerging pattern underlying a higher stroke severity specifically in women.

Previous **Figure 5**:

Updated Figure 5:

Any further changes and point-by-point responses to all concerns raised by the referees are outlined in the following. We believe the updated manuscript is significantly stronger as a result. Our comments are in *italics* and reviewer queries are in **bold**. New material is in red in the revised manuscript, as well as in this revision letter. To ensure a better overview, only a few examples of new or updated figures are reproduced in this response letter, while they are displayed in full in the revised manuscript and supplementary materials documents. Updated code can be found openly available online: https://github.com/AnnaBonkhoff/BMH_stroke_severity_sex_differences.

Reviewer #1 (Remarks to the Author):

The present study reports a combined lesion symptom mapping (LSM) and Bayesian analysis conducted in 555 patients (and replicated in other 503 patients) in order to explore male and female-specific lesion patterns underlying stroke severity as assessed with NIHSS. This analysis was able (1) to uncover a more widespread pattern of relevant lesions in women than in men, (2) to emphasize the importance of lesions in the left hemisphere PCA territory in generating higher stroke severity in women and (3) to suggest sex-specific functional cerebral asymmetries to be further used in acute stroke therapies. This study focuses on an interesting topic seldom investigated explicitly and therefore could be of significance to the field.

We are grateful for the referee's positive evaluation of our work.

Unfortunately, several points are difficult to follow (also because of the novelty of the method) and should be more clearly or explicitly stated:

1.1) The method seems interesting, but not enough details are provided and the reader is referred to another paper not peer reviewed yet...

We would like to thank the referee for their time and effort reviewing our manuscript and we now provide more in-depth descriptions, justifications and contextualization of our approach to allow other laboratories to understand and reproduce our analysis workflow.

*We, first of all, deleted direct references to preprints, for example, **Results (p. 7):***

*~~"As in previous work,~~ We successively combined 1) the automated low-dimensional embedding of high-dimensional **DWI-derived** lesion information via non-negative matrix factorization (NMF)¹, and 2) probabilistic modeling, **based on the latent NMF embedding**, to simulate the prediction of acute stroke severity, as measured by the National Institute of Health Stroke Scale (NIHSS)."*

Furthermore, we added comprehensive descriptions of the non-negative matrix factorization algorithm and the motivation behind this choice of factorization algorithm:

Methods, Derivation of a low-dimensional lesion representation (p. 24):

*"To extract reoccurring, archetypical **and directly interpretable** lesion patterns in our stroke population **in a data-driven, unsupervised fashion**, we performed NMF¹ on the log-transformed brain region- and tract-wise lesion load. By these means, we obtained ten **unique topographical lesion configurations**, that we will call *lesion atoms* in the following (c.f., **Supplementary Materials** for further details)."*

Supplementary Materials, Unsupervised lesion embedding derivation (p. 1):

*"We employed non-negative matrix factorization (NMF) to compute a low-rank approximation of our lesion data: Let the matrix **V** represent the region-wise lesion loads for all subjects and thus be of $p \times n$ dimensions (p = number of grey matter regions/white matter tracts and n = number of subjects). NMF decomposes the matrix **V** into a latent factor representation **W** and a latent factor loading **H**, i.e., $V=WH$ with $p \times k$ and $k \times n$ dimensions (k =number of latent spatial patterns).¹ The hidden factor representation **W** then links the derived low-rank lesion embedding to the lesion information in the original anatomical brain regions. The matrix of hidden factor loadings **H** assigns relevances to each specific lesion pattern within the lesion embedding to characterize individual patients' actual lesions. We opted for this multi-to-multi mapping representation, as we expected to derive biologically more meaningful stroke patterns in view of NMF's positivity constraint. This important quality of NMF stands in contrasts to alternative matrix factorizations algorithms, such as principal component analysis (PCA): When using this alternative decomposition approach, an individual patient's lesion would have been retrieved through inscrutable additions and subtractions of low-dimensional lesion pattern, which would have hindered an intuitive and direct interpretation of lesion pattern effects."*

Figure caption 1 (p. 39):

“A data-driven pattern-discovery framework enabled the derivation of coherent patterns of stroke lesion topographies directly from the segmented high-resolution brain scans from 555 stroke patients. This unsupervised, multi-to-multi mapping approach led to unique, predominantly right-hemispheric (A) or left-hemispheric (B) lesion patterns. [...] The anatomical plausibility of our derived lesion pattern may particularly stem from the positivity constraint of the non-negative matrix factorization algorithm, an advantageous quality that motivated our choice of dimensionality reduction technique. Conversely, alternative matrix factorization algorithms, such as principal component analysis (PCA), would have hampered a straight-forward interpretation of lesion pattern effects by encoding individual lesions through more incomprehensible additions and subtractions of low-dimensional lesion pattern.”

1.2) For instance, it is not clear how authors dealt with well-known LSM problems such as the impact of vascular territories in lesions localization, or data correlations among neighboring voxels (see also Mah et al., 2014; Sperber, 2020; Toba et al., 2020; Vaidya et al., 2019), to mention just a few.

Also, how this method is situated compared with multivariate methods that begin to be used in the field (e.g., Smith et al., 2013; Zhang et al., 2014; Yourganov et al., 2016; Toba et al., 2017; Pustina et al., 2017)? All these details should be mentioned in the manuscript so that the reader can understand the method without referring to other preprints.

We thank the referee for having raised these points, in response to which we have now extended the discussion of potential limitations of our presented approach, as well as the discussion of the pros and cons of uni- and multivariate lesion-symptom approaches substantially:

Methods, Explaining inter-individual differences in acute stroke outcomes (p. 26):

“Main advantages of our analytical approach may be seen in i) the enhanced interpretability of estimated NMF-components, i.e., *lesion atoms*, ii) the multivariate nature of the stroke lesion embedding, and iii) the possibility of integrating interaction effects between lesion atoms and an individual’s sex in a statistically sound probabilistic modeling framework. We thus make a step towards providing an alternative approach to classical voxel-wise lesion-symptom mapping (VLSM) analyses,² that estimate the magnitude of effects in one brain location at a time and thus fit one model per voxel. Our approach opens an alternative window to “one voxel at a time” by fitting a *single* model to the entire brain. All ten of our *lesion atoms* were entered into the regression model at once. Hence, we could estimate the effect of each *lesion atom*, while accounting for the effects of all further *lesion atoms*. By these means, our approach resembles more machine-learning-based multivariate lesion-symptom mapping methods, that have been introduced recently to evaluate lesion patterns instead of single voxels in association to an outcome.³ Examples can be seen in approaches leveraging support vector machines,^{4,5,6} tree-based⁷ or game theory-based algorithms.⁸ Importantly, all of these multivariate approaches may decrease the distortion of functional localization compared to univariate voxel-wise approaches by considering subcortically and cortically located lesions at once.^{9,10} We additionally aimed to mitigate confounding effects of excessively correlated neighboring voxels, as naturally arising due to the stroke lesion-vasculature dependence, by explicitly combining frequently jointly affected brain regions in lesion patterns, i.e.,

our *lesion atoms*. Despite these methodological innovations, no currently existing stroke modeling approach may achieve unailing causal conclusions.^{9”}

Discussion, Limitations and future directions (p. 19):

“Similarly, the spatial resolution of our Bayesian hierarchical approach stays within the realms of frequently arising, typical lesion patterns and back-transformed atlas brain regions. Consequently, our region-wise approach does not allow for a comparably high spatial resolution of lesion-symptom analyses relying on voxel-wise data.^{2,4} Nonetheless, this reduction in lesion dimensionality was necessary to render sex-aware analyses within our Bayesian hierarchical model framework feasible. [...] What is more, our Bayesian models could, for example, be modified to integrate an additional hierarchical level to differentiate between rich-club and non-rich-club regions^{11,12} or contrast agglomerated and separately modeled sets of brain regions to test *brain modes*, as introduced by Godefroy and colleagues in 1998.^{13,14”}

1.3) Authors specify that lesion atoms 6 and 10 denoted the vascular supply territory of the posterior circulation. This can be an artifact! Please explain how the role of the vascular territories was controlled.

We agree with the referee that the method of assigning roles of vascular territories to lesion atoms requires additional clarification. Vascular territories were, indeed, assigned post-hoc, via expert-based evaluations by several board-certified neurologists with clinical expertise in vascular neurology and neuroimaging analysis of cerebrovascular disease. We now describe this process more in detail and are generally more careful in expressing links to the posterior circulation.

Methods, Derivation of a low-dimensional lesion representation (p. 24):

“Probable vascular territories of *lesion atoms* were assigned via evaluations by individuals with several years of neurology and neuroradiology experience (A.K.B., M.B., N.R.).”

Exemplarily here: Abstract (p. 4):

“Furthermore, particular lesions in the presumed posterior circulation of the left hemisphere consistently underlay a higher stroke severity exclusively in women. These sex-sensitive lesion pattern effects were ~~discovered and subsequently~~ robustly validated in ~~two~~ a second large independent, multisite dataset (n=503, 41% female).”

Here: Results, Validation analyses (p. 11):

“In particular, we found substantial differences between men and women in a *lesion atom* that predominantly comprised left-hemispheric, presumably posterior cerebral artery-supplied regions (*lesion atom 10*: difference distribution: mean=-2.68, HPDI=-4.92- -0.733 (i.e., no overlap with zero)).”

Or here: Discussion (p. 14):

“These sex discrepancies were particularly pronounced for lesions affecting a number of left-hemispheric regions, i.e., thalamus, hippocampus and occipital cortical brain regions, thus regions that are reminiscent of the vascular territory of the posterior circulation.”

1.4) It is not clear either whether authors considered variables such as cerebral atrophy known to play a role in post-stroke disorders (see Rothwell et al., 2005) such as those assessed with NIHSS.

The referee is highlighting an important point regarding the effect of pre-existing age-related cerebral changes, such as cerebral atrophy. However, capturing these changes veridically usually requires serial MRI scans, which were not available in the cross-sectional cohorts of stroke patients included in these analyses.

While we thus did not have access to any direct quantitative measures of cerebral atrophies, we ensured to address potential confounds severalfold: Firstly, we spatially normalized all images and lesion masks to a common reference space, thereby compensating for varying brain sizes. Every single image and lesion mask pair, of both the derivation and validation cohort, was subsequently rigorously quality controlled to ensure accurate representations of lesioned brain tissue. Secondly, we incorporated both age and non-linear interactions of age (age^2) as covariates in all of our analyses. Moreover, we consistently included covariates, such as the white matter hyperintensity lesion load, that can be regarded as proxies of chronic brain health and correlate with cerebral atrophy.¹⁵ Lastly, we observed female-specific effects in samples of men and women that did not differ in age (e.g., the validation cohort and the non-cardioembolic stroke stratum) and should thus be affected by similarly strong age effects.

We have inserted the following additions to transparently present these considerations to the reader:

Methods, Magnetic resonance imaging: Preprocessing (p. 23):

“This preprocessing pipeline, especially featuring the co-registration of DWI and FLAIR images, was optimized to generate as reliable and accurate spatial normalizations for as many subjects as possible. The quality of normalized lesion masks was carefully inspected by two experienced raters (A.K.B, M.B).”

Methods, Explaining inter-individual differences in acute stroke outcomes (p. 24):

“In view of women’s overall higher age, we aimed to adjust for age-specific effects particularly comprehensively by not only taking into account the basic age, yet also its squared value.”

Discussion, Limitations and future directions (p. 19):

“Lastly, we did carefully account for inter-individual differences in important sociodemographic characteristics (e.g., age), comorbidities (e.g., atrial fibrillation), markers of chronic brain health (e.g., white matter hyperintensity lesion load) and stroke subtypes (e.g., cardioembolic stroke genesis). Additionally, each image and corresponding lesion mask was individually quality controlled to ensure the absence of overt spatial distortions (e.g., due to cerebral atrophies).”

2) Authors should better emphasize that analyses were conducted in two separated (derivation and replication) groups and that results were similar, although some differences could be observed. For instance, how do the authors explain that in the replication analyses, 5 lesion atoms have been found in the right hemisphere, while only 4 have been reported for the derivation group? This is an important

issue because it shows the instability of behavioural and lesional patterns even when analyses are conducted in more than 500 patients. This information should also be present in the abstract.

We thank the referee for having motivated a more focused evaluation and presentation of derivation and validation cohort results. While doing so, we have detected the previously outlined inconsistencies in lesion data representation (c.f., beginning of this response letter).

Upon re-analyzing the derivation cohort's data with harmonized lesion data input, the non-negative matrix factorization (NMF)-step led to 5 predominantly right- and 5 left-hemispheric lesion atoms – similar to the ratio in the validation cohort.

In addition to correlating NMF-components of derivation and validation cohort, as already presented in our initial manuscript version, we now also conducted new analyses in which we expressed each cohort's lesion data by employing the lesion embedding transformation derived from the other cohort's lesion data.

Overall, NMF-components of both cohorts correlated significantly, and results remained highly similar after utilizing the other cohort's embedding. Therefore, we feel confident that our pipeline generates stable results at given sample sizes. Nonetheless, we do acknowledge that subtle differences still exist despite ~500 subjects, which we now more accurately express in the updated manuscript version.

As suggested by the referee, we now more explicitly report that results originate from a derivation cohort with 555 subjects and a validation cohort with 503 subjects:

Abstract (p. 4):

“This framework was tailored to carefully estimate possible sex differences in lesion patterns explaining acute stroke severity (NIHSS) in 555 patients (38% female). [...] These sex-sensitive lesion pattern effects were ~~discovered and subsequently~~ robustly validated in ~~two~~ a second large independent, multisite dataset (n=503, 41% female).”

Introduction (p. 7):

“For this purpose, we leveraged neuroimaging data originating from two large, independent hospital-based cohorts gathering data of 555 (derivation) and 503 (validation) acute ischemic stroke (AIS) patients in total.”

The aforementioned additional analyses are outlined in the methods and results sections:

Methods (p. 27):

“To further ascertain the robustness of our unsupervised lesion embedding, we correlated NMF-components as computed for the derivation and validation cohort. Moreover, we re-computed region-wise relevances in final non-hierarchical analyses based on the other cohort's lesion embedding (i.e., we first estimated the NMF-transformation based on one cohort's imaging data and then applied this transformation to the data of the other cohort). We then ran additional correlation analyses between region-wise relevances as originating from the cohort-specific embedding, as well as the other cohort's embedding to ensure our results were not substantially altered by the particular embedding.”

Results, Validation analyses (p. 11):

“While there were subtle differences of lesion embeddings, likely expressing sample-specific lesion distributions, *lesion atoms* of the derivation and validation dataset were overall highly correlated. This high correlation indicated that our unsupervised approach facilitated the derivation of similar lesion topography embeddings in both independent datasets (Supplementary Table 4). Additionally, back-projected region-wise relevances were highly correlated when computed based on the other cohort’s lesion embedding (derivation cohort: $r=0.84$, $p<0.001$; validation cohort: $r=0.78$, $p<0.001$). Both of these correlation analyses combined thus highlighted the distillation of largely similar archetypical lesion pattern in both cohorts and the independence of results from the concrete lesion embedding.”

Furthermore, we additionally point to differences in derived lesion embeddings and the possibility to circumvent those in future research in the discussion section:

Discussion, Limitations and future directions (p. 19):

“We here employed two frequently used brain atlas templates^{16,17} and unsupervised dimensionality reduction.¹ Future studies could explore differing atlas templates and also supervised strategies to organize brain regions. Such supervised approaches could reduce variability between lesion embeddings of different datasets, since we here saw subtle differences in automatically derived *lesion atoms*. Eventually, confidence in our results could then be increased in case of successful replication despite these methodological alterations.”

3) Authors used both Harvard Oxford and JHU atlases, but they do not explain how they dealt with overlapping voxels. This is an issue with important consequences on statistical analyses. Why did the authors not use a template integrating both grey and white matter?

We acknowledge this point and would like to thank the referee for prompting further justifications for the choice of templates. Both atlases are widely used in the brain-imaging community for parcel-wise averaging of measurements – thus we chose to conform to this established approach in the present study.

More specifically, we decided to employ the Harvard-Oxford (HO) cortical and subcortical and John-Hopkins University (JHU) white matter tract templates for several reasons, which we will outline in the following: First of all, our aim was to choose well validated and frequently employed templates. In view of >1.000 citations for each of the publications linked to both templates, we felt that these requirements were met (HO: Desikan et al., 2006: 6.684 citations; Wakana et al., 2007: 1.344 citations, as of Dec 5, 2020). We had furthermore relied upon the grey matter regions of HO template in an earlier project¹⁸ and strived to ensure maximum consistency and comparability between projects, while however, optimizing the previous approach by now considering white matter regions as well. Moreover, we assumed that only few lesioned voxels would be captured by both the HO and JHU template. Indeed, this was the case for only 1.6% of all lesioned voxels, mostly located in bilateral thalamus and anterior thalamic radiation. Within the framework of this revision, we repeated analyses after excluding any overlapping voxels for the white matter tracts. Results remained the same (correlation of NMF-components before and after excluding overlapping voxels: $r=0.88$, $p<<0.001$, correlation of back-projected region-wise results: $r=0.86$, $p<<0.001$).

Lastly, to the best of our knowledge, no template exists that comprises high-granular information on both grey matter regions, as well as white matter tracts – likely due to differing imaging modalities during region of interest extraction (T1-weighted images for grey matter regions vs. diffusion tensor imaging (DTI) sequences for white matter tracts).

We now discuss the effect of employed templates and the need to replicate findings when relying on different ones in the discussion section:

Discussion, Limitations and future directions (p. 19):

“We here employed two frequently used brain atlas templates^{16,17} and unsupervised dimensionality reduction.¹ Future studies could explore differing atlas templates and also supervised strategies to organize brain regions. Such supervised approaches could reduce variability between lesion embeddings of different datasets, since we here saw subtle differences in automatically derived *lesion atoms*. Eventually, confidence in our results could then be increased in case of successful replication despite these methodological alterations.”

4) For the second (replication) group, only scarce information is provided (age: 65.0 (14.6), sex: 40.6% female, NIHSS: 5.48 (5.35)). The NIHSS score provided here represents the mean, or the median (as provided for the derivation group in Table 1)? Please include for the replication group a table with the same information as for the derivation group, so that the reader can compare data.

We appreciate the feedback regarding the reporting of clinical information on the second cohort. We have now expanded the cohort description in the methods section and added a table containing information on the exact same variables as stated for the derivation cohort. We hope this addition facilitates further independent comparisons of cohorts.

Methods (p. 27):

“(mean age(SD): 65.0 (14.6), sex: 40.6% female, mean NIHSS(SD): 5.48 (5.35)) originating from five centers constituted the finally included sample (c.f., **Supplementary Table 1** for further clinical characteristics).”

Supplementary Materials (p. 3):

“**Supplementary Table 1. Patient characteristics of the validation cohort.** Mean (SD) unless otherwise noted. The groups of male and female patients were compared via two-sample t-tests or Fisher’s exact tests as appropriate. Asterisks indicate significant differences between men and women.”

	All participants (n=503)	Women (n=204)	Men (n=299)	
Age	65.0(14.6)	65.3(16.3)	64.8(13.2)	p=0.70
Sex	59% male,41% female	-	-	
NIHSS	5.5(5.4) (median(iqr): 4(5))	5.8(5.6) (median(iqr): 4(6))	5.3(5.2)	p=0.31

			(median(iqr): 4(5))	
Normalized DWI-derived stroke lesion volume (ml)	21.0(40.5) (median(iqr): 3.9(19.8))	22.9(38.7) (median(iqr): 4.0(31.3))	19.7(41.7) (median(iqr): 3.7(15.0))	$p=0.38$
White matter hyperintensity lesion volume (ml)	11.13(13.6) (median(iqr): 5.8(12.3))	9.6(11.6) (median(iqr): 4.9(13.2))	12.2(14.8) (median(iqr): 6.2(12.4))	$p=0.04^*$
Hypertension	63.6%	67.2%	61.2%	$p=0.19$
Diabetes mellitus type 2	22.7%	20.6%	24.1%	$p=0.39$
Atrial fibrillation	19.3%	24.0%	16.1%	$p=0.03^*$
Coronary artery disease	16.9%	11.8%	20.4%	$p=0.01^*$

5) An important part of the NIHSS score is represented by motricity. Authors should better emphasize this point and precise that current lesion atoms are biased by the use of such a scarce scale to evaluate behaviour.

*We fully agree with the referee that the NIHSS-based stroke severity that we evaluated is dominated by motor symptoms (and language impairments), as the updated manuscript states in **the Discussion (p. 14)**:*

“This distribution of main weights was rendered particularly plausible, given that the NIHSS scale assigns the majority of its points to motor and language functions.”

*To emphasize this aspect even further as suggested by this referee, we have amended the **Discussion, Limitations and future direction paragraph (p. 19)**:*

*“Furthermore, we here explored sex disparities in lesion patterns of *global* stroke severity, which allows for some broad clinical implications of our findings. **Naturally, our results may, to a certain extent, be dominated by effects due to motor symptoms, given their disproportionately high importance for the overall NIH Stroke Scale score.** To evaluate more specific brain functions at acute and chronic stages, future studies could **therefore** focus on sub-items of the NIHSS, **such as language impairments, dysarthria, disturbances in orientation, neglect or on specific cognitive behavioral tests, e.g., probing memory functions.**”*

6) It is not clear why authors did correct for the sex covariate in the first analyses (line 219), and considered it in the model only in a second step.

We would like to thank the referee for this feedback and now added a more precise description of in what way and why we integrated information a patient’s sex as independent input variable. Essentially, our motivation in inserting a separate input variable for each patient’s sex (i.e., in all, also the non-hierarchical, first analyses) in addition to estimating male- and female-specific lesion pattern effects (i.e., in the second, hierarchical and sex-specific analyses, the second step), was exactly to disentangle sex differences

dependent and independent from specific lesion patterns. We clarified our modelling choices in the revised manuscript version in the following way:

Methods, Explaining inter-individual differences in acute stroke outcomes (p. 24):

“Importantly, we **also** included sex as a **non-hierarchical, lower-level** variable in the model to capture sex differences in stroke severity that were *independent* of lesion patterns. **Women may, for example, also present with more severe strokes on average. Such a sex-specific, but lesion pattern-independent difference in stroke severity could then be modeled by the sex variable. It would, however, not have an influence on lesion pattern-sex interaction effects. In other words: The non-hierarchical sex variable represents the “residual” sex effects on the outcome that are not already captured by the hierarchical sex-specific NMF components.**”

Results (p. 9):

“Notably, we included sex as a variable in the model to differentiate between sex differences in stroke severity that were *dependent* and *independent* of lesion patterns. If, for example, stroke severity was generally higher in women, without any link to the actual lesion pattern, this effect would be represented by the Bayesian posterior distribution of this sex variable, but not in the sex-specific *lesion atom* Bayesian posterior distribution. In contrast, sex-specific *lesion atom* distributions would indicate interaction effects on the outcome, i.e., effects that were specific to an individual’s sex *and* the precise *lesion atom*.”

7) In the first group, age was significantly different between men and women. Can this element have an impact on potential cerebral atrophies for instance? How this issue has been controlled in the analysis?

We agree with the referee that age-related effects are crucial to consider, given that stroke populations are inherently aging populations in general and men and women of our derivation cohort may differ in age in particular.

We would like to primarily refer to our response to question 1.4, where we give an in-depth description on how we addressed potential age and age-related effects, such as cerebral atrophies. Additionally, we would like to emphasize that female-specific effects also arose for men-women samples that did not significantly differ in age (e.g., men and women of the validation cohort or men and women in the stratum of non-cardioembolic stroke patients). This circumstance renders more pronounced cerebral atrophies in women as origin of our observed sex differences improbable.

*Nevertheless, the more advanced age in women in our derivation cohort combined with our ancillary analysis on sex-age interactions – highlighting more pronounced female-specific effects for patients above 52 years of age – could explain a part of why we generally detected more substantially altered lesion atoms in our derivation cohort compared to our validation cohort. We make these considerations explicit to future readers of our work in **Results, Ancillary analyses (p. 13):***

“Further, this observable sex-age interaction might also explain why we discovered more extensive female-specific effects in the derivation cohort, featuring significantly older female

patients, than in the validation cohort that was characterized by a non-significant age difference between men and women.”

8) Why is an uncorrected p provided in supplementary tables 1 and 2, while in the text it is mentioned that analyses are Bonferroni corrected (line 196)? Please be coherent.

We thank the referee for pointing out this inconsistency in evaluating and displaying results of region-wise lesion load comparisons. We now harmonized reports in the results section and supplementary tables, i.e., supplementary tables now show the Bonferroni-corrected (and altogether non-significant) p-values that we refer to in the results section (supplementary tables are not reproduced here to ensure a better overview, yet can be found in the updated supplementary materials file).

9) Discussion could be further developed to integrate the translational neuroscience aspect of these findings and particularly the personalized clinical rehabilitation.

*We are grateful for the referee’s suggestion to expand on the translational neuroscience implications our findings may have. We have extended the respective section in the **Discussion section (p. 18)**:*

“Most previous studies have relied on sex-independent, general cut-offs of salvageable tissue volumes to decide, for example, whether to undertake mechanical thrombectomy in the later time window.¹⁹ **A conceivable first step could be to revisit previous randomized clinical trials’ data to investigate rescued-tissue-sex interactions.** Future clinical treatment studies could **then prospectively and** systematically test varying cutoffs for men and women, hypothesizing that rescuing a lesser amount of tissue in women would still be sufficient for a noticeable positive treatment response. [...] **Notably, such studies should also consider additional age-sex interactions, given that our ancillary analyses suggested more pronounced female-specific effects in case of more advanced age.**”

*As well as in **Discussion, Limitations and future directions (p. 19)**:*

“Above all, it may be especially fruitful to examine whether sex differences observed here generalize to cerebral reorganization processes during the recovery phase post-stroke. In the positive case, generated results could fuel personalized clinical rehabilitation endeavors.²⁰”

Minor:

10) Please be consistent with terms used for the first group. Currently both ‘derivation’ and ‘development’ terms are used.

We thank this referee for the feedback and now ensure that the first cohort is consistently called “derivation cohort” throughout the entire manuscript. Likewise, we harmonized the term used for the second cohort and now refer to the “validation cohort” throughout the entire manuscript.

11) The title of the fig 3 can be misleading ‘Sex-specific posterior distributions’. Please consider to define the term ‘posterior’ here, that could be interpreted as related to posterior anatomical territories.

We would like to thank the referee for drawing attention to the ambiguous use of the term “posterior”. We now ensure to refer to “Bayesian posterior” in all instances in which we refer to the mathematical Bayesian posterior distribution, rather than the anatomical posterior circulation, e.g., Figure 3 (p. 41):

“Figure 3. Sex-specific Bayesian posterior distributions of all ten lesion atoms and overall whole-brain region-wise relevance to explain stroke severity in women (left) and men (right).”

Further example: Results, Lesion atom and regional relevance for stroke severity (p. 9):

“Out of the ten derived lesion atoms, five atoms possessed a substantial explanatory relevance for acute stroke severity. This relevance was inferable from Bayesian posterior distributions of lesion atoms that did not substantially overlap with zero.”

12) Please correct the reference 23 in 2,3 (line 131) and ‘globus pallidum’ (line 230)

We thank the referee for having spotted these citation and spelling errors, which we corrected in the revised manuscript version.

Reviewer #2 (Remarks to the Author):

Sex-specific lesion topographies explain outcomes after acute ischemic stroke

Anna K. Bonkhoff et al. 2020

Peer Review for Nature Communications

This a compelling article on gender-specific functional cerebral asymmetries and gender-stratified deficit after acute ischemic stroke. The authors test gender differences in lesion patterns that explain stroke severity in two large independent datasets. They find that females show a more widespread pattern of lesions explaining acute severity. They argue that lesions in the left hemisphere's posterior circulation relate to a higher stroke severity exclusively in women. According to the authors, the findings suggest gender-specific functional cerebral asymmetries, which would motivate a more gender-sensitive treatment decision approach.

The study addresses a potentially high impact issue in the comprehension of gender differences in the prediction of stroke severity that may contribute to the development of gender-specific stroke therapies. The manuscript is well written, the cohort is adequate, and the methodology is potentially proper, but several significant caveats require attention. Some of these issues may not be solvable, which could impact the main message.

We thank this referee for the very positive assessment of our work.

Major critiques

1) The stroke dataset may not be representative of a clinically relevant sample. The NIHSS scores are overall low (mean 3), which is not typical of a prospective sample where the mean/median is around 7. The Authors should discuss inclusion/exclusion criteria and the presence of possible biases of selection. The authors do not specify the exact timing of NIHSS score recordings vis-a-vis the onset of stroke.

"Within 48h from admission" is not acceptable, primarily as they do not provide any information on acute revascularization treatment of these patients (e.g., is the NIHSS before or after treatment?). The NIHSS scores can enormously change in the first hours from stroke onset depending on various factors, including micro-embolic signals, spontaneous reperfusion, blood pressure and perfusion pressure, metabolic homeostasis, brain edema, or seizure, etc., and these changes differ according to etiology (see next point).

We are grateful to the referee for motivating a more in-depth discussion of our sample characteristics. We agree that these considerations are particularly essential when extrapolating the implications of our study's findings.

*With respect to stroke severity: Our derivation and validation cohort's median NIHSS scores (3 and 4, respectively) match those scores reported for large unselected stroke registry samples of patients surviving the acute phase (e.g., Smith et al., 2010, Get With the Guidelines Stroke-Program data: median NIHSS: 4²¹; Gattlinger et al., 2019, Austrian Stroke Unit Registry: median NIHSS: 3²²). Naturally, these average scores are lower than in randomized clinical trials that typically employ a minimum NIHSS cut-off-score for inclusion. While we therefore think that our stroke severity is approximatively representative of an unselected stroke sample, we do acknowledge that we nonetheless may have undersampled severely affected, elderly women (c.f., **Minor 11** for further details and respective changes in the revised manuscript version).*

With respect to the exact timing of data recordings and acute revascularization treatment: Inclusion criteria required an imaging data acquisition during the first 48 hours after admission only. This time window was crucial for the large-scale recruitment of stroke patients and essentially, to allow for the feasibility of conducting the study in clinical routine. The acute revascularization treatment was recorded for a subgroup of patients only, which is why we decided against considering it as covariate; especially since we did not have information on whether the treatment was administered before or after the image acquisition. We now clearly acknowledge these known limitations of our observational study design (c.f. below), but we feel reassured by the verification of our main findings in an independent, similarly derived hospital-based cohort of patients.

Motivated by the referee's comment, we also aimed to indirectly address these aspects by means of a subgroup analyses that we outline in the following.

Methods, Ancillary analyses (p. 27):

"We performed three ancillary hierarchical analyses. Firstly, we aimed to reduce potential confounds arising from dispersed time points of data acquisition and effects of acute recanalization therapies and limited the derivation cohort sample to only those subjects whose data was acquired directly upon admission (and not within the first 48 hours, as for the entire cohort). Thus, their data acquisition likely preceded the potential administration of any acute recanalization therapy and their lesion-symptom associations may be considered to be purer."

Results, Ancillary analyses (p. 12):

"While data were acquired within the first 48 hours after admission for the entire derivation cohort, we aimed to reduce potential confounds due to varying times of imaging and stroke severity acquisition, as well as any acute revascularization therapy effects by specifically

investigating those patients with MRI imaging and NIHSS score available upon their admission. These criteria were fulfilled by 152 patients in total (men: 87, mean age(SD): 63.1(12.3), mean NIHSS(SD): 4.6(5.6); women: 65, mean age(SD): 69.6(14.5), mean NIHSS(SD): 5.5(6.5); two-sided t-tests: age: $p=0.003$, NIHSS: $p=0.45$, **Supplementary Figure 6**). The Bayesian posterior difference distribution of *lesion atom* 10, representing presumed left PCA-supplied brain regions, indicated a substantial female-specific effect in explaining a higher stroke severity (**Supplementary Figure 7**).”

New Supplementary Figures 6 & 7 can be found in the updated supplementary materials document.

*Lastly, we now added this limitation in our **Discussion, Limitations and future directions section (p. 20)**:*

“[...] Also, we did not have access to some measures, which could be of potential interest as covariate in our analysis, for a majority of patients; for example, delays in hospital arrival, the exact time between imaging and NIHSS acquisition, acute changes in stroke severity (e.g., due to spontaneous reperfusion, brain edema or seizures) or administered revascularization therapies. In this study, we aimed to address these factors indirectly by tailoring ancillary analyses to subgroups, e.g., only those patients whose data was acquired directly after admission. Future studies could recruit equal numbers of men and women, investigate these further covariates more explicitly and additionally consider interaction effects of any of these variables with the patient’s sex status.”

2) Women are known to go later to the ER, which could bias the results: was this data collected?

*We agree with the referee that female-specific greater delays in hospital admission are an important sex difference – which we already featured in our **Introduction (p. 5)**: “Further sex differences relate to women [...] having a higher risk of delays in hospital arrival.”), but now acknowledge and address even more explicitly.*

Our data was assembled within the first 48 hours after admission for both men and women and thus, for the most part, not in the first few hours after symptom onset. Therefore, we assumed that the variability in the time from symptom onset to hospital arrival is small compared to the variability in time from symptom onset to data acquisition. Hence, we feel it is reasonable to assume that sex-specific delays in hospital arrival may not have a substantial effect given our specific sample.

*Nonetheless, we agree that future studies, that have access to this information, are warranted, which we now express in **Discussion, Limitations and future directions (p. 20)**:*

“[...] Also, we did not have access to some measures, which could be of potential interest as covariate in our analysis, for a majority of patients; for example, delays in hospital arrival, the exact time between imaging and NIHSS acquisition, acute changes in stroke severity (e.g., due to spontaneous reperfusion, brain edema or seizures) or administered revascularization therapies. In this study, we aimed to address these factors indirectly by tailoring ancillary analyses to subgroups, e.g., only those patients whose data was acquired directly after admission. Future studies could recruit equal numbers of men and women, investigate these further covariates more explicitly [...]”

3) The etiology of stroke is missing. The prevalence of different etiologies is different between men and women (i.e., cardioembolic vs. other causes), which would affect the anatomical distribution of lesions.

We would like to thank this referee for motivating a more thorough investigation of stroke subtype effects. While we had previously included the covariate “atrial fibrillation” as a proxy of cardioembolic stroke genesis in our model, we have now conducted novel ancillary analyses to address the role of cardioembolic strokes and associated lesion patterns more comprehensively.

*We describe how we approached this analysis in **Methods, Ancillary analyses (p. 27)**:*

“Conceivably, female-specific lesion effects could be due to a higher frequency of cardioembolic strokes and associated typical lesion patterns (e.g., more multifocal stroke lesions).²³ In a second ancillary analysis, we therefore sought to disentangle stroke-subtype- and sex-specific effects and concurrently performed stroke-subtype- and sex-aware analyses, featuring two hierarchical levels. Accordingly, patients were not only stratified into a group of men and women as in main analyses, yet initially assigned to a group of either cardioembolic or non-cardioembolic stroke genesis and then split into men and women (c.f., **Supplementary Materials** for a comprehensive display of model specifications). We then contrasted male- and female-specific lesion pattern effects within the cardioembolic and non-cardioembolic stroke subgroups. To allow for a reasonably high number of patients in each of these four subgroups, we here merged data of the derivation and validation cohorts (c.f., **Supplementary Materials** for further details).”

*Our findings in **Results, Ancillary analyses (p. 12)**:*

“Women, especially if older, are known to more frequently experience atrial fibrillation and cardioembolic strokes.²⁴ The motivation of our second ancillary analysis was to explicitly exclude the possibility that observed sex differences in explanatory lesion patterns merely stemmed from varying frequencies of cardioembolic strokes. As information on stroke subtype was available for subgroups of each cohort only, we here pooled data from the derivation and validation cohort to rest analyses on a sample as large as possible. Approximately one fourth of the 880 patients in total experienced a cardioembolic stroke (203 patients with cardioembolic stroke: men: 108, mean age(SD): 66.6(14.3), mean NIHSS(SD): 6.3(5.4), women: 95, mean age(SD): 74.3(12.9), mean NIHSS(SD): 7.5(7.0); two-sided t-tests: age: $p < 0.001$, NIHSS: $p = 0.17$; 677 patients with non-cardioembolic stroke: men: 419, mean age(SD): 62.8(13.3), mean NIHSS(SD): 4.7(5.3), women: 258, mean age(SD): 63.7(16.4), mean NIHSS(SD): 5.1(5.7); two-sided t-tests: age: $p = 0.45$, NIHSS: $p = 0.44$, **Supplementary Figure 8**). As expected, women featured a higher frequency of cardioembolic strokes (Fisher’s exact test: $p = 0.03$). We detected substantial female-specific *lesion atom* effects independent of whether comparing men and women with cardioembolic strokes, or men and women with non-cardioembolic stroke genesis: in case of cardioembolic stroke, difference distributions of *lesion atoms* 4, 6 and 8 suggested a higher explanatory relevance exclusively in women (**Supplementary Figure 9**), while *lesion atoms* 1, 2, 9 and 10 had a substantially higher relevance in women in case of non-cardioembolic stroke (**Supplementary Figure 10**). Since we extracted female-specific effects in strata of both only cardioembolic and non-cardioembolic stroke patients, these results rendered the interpretation of lesion pattern effect differences due to varying frequencies of cardioembolic stroke unlikely. Furthermore, all of these female-specific effects emerged for samples that were fairly even in their numbers of men and women (c.f., cardioembolic stroke: 108 men and 95 women), or did not comprise any

significant differences in age and stroke severity (c.f., non-cardioembolic stroke: two-sided t-tests: age: $p=0.45$, NIHSS: $p=0.44$), which additionally increased the confidence that sex differences did not artificially arise from these differences.”

*And additional aspects in **Supplementary Materials, Ancillary analyses: Stratifying for cardioembolic stroke genesis (p. 8)**:*

“Data of the derivation and validation cohorts were merged in ancillary analyses focused on the stroke subtype and age of participants. More specifically, we adopted the NMF-transformation as learned exclusively based on the derivation cohort and applied the same transformation to the validation cohort. To harmonize data of both cohorts further, we normalized NMF-transformed lesion data before merging and inserting it into the linear regression model. We refrained from re-computing the NMF-transformation based on the merged data and chose to rely on the derivations cohort’s lesion embedding instead to increase the ease of interpretation: In this way, any results could immediately be compared to the derivation cohort’s results. This choice was furthermore supported by our analyses demonstrating the independence of the concrete lesion embedding (c.f., **Results: Validation analyses**). Lastly, we added cohort membership as a covariate to the model to further account for differences between cohorts.”

Newly added supplementary figures, as well as the full model specifications can be found in the revised supplementary materials document.

In view of these results, we thus feel confident to conclude that female-specific lesion atom effects arose independent from cardioembolic or non-cardioembolic stroke subtypes. Indeed, the higher relevance of lesion atom 10 in women, the most reliably detectable effect, emerged in the subgroup of non-cardioembolic stroke patients and was thus not depending on a cardioembolic stroke-related stroke lesion pattern.

4) It is not clear whether stroke lesions were focal or multifocal in each subgroup. This information typically relates to differences in stroke etiology --multifocal lesions are embolic--and could also impact stroke severity compared to focal lesions.

*We thank the referee for raising this important point. While we did not extract information on the focal or multifocal nature of stroke lesions, we aimed to account for more frequent focal vs. more frequent multifocal lesions in our novel ancillary analyses focused on cardioembolic vs. non-cardioembolic stroke subtypes. We observed female-specific more substantial lesion atom effects independent of the stroke subtype, which rendered the interpretation that sex differences arose from systematic lesion pattern unlikely. We refer to **question 3** for details of this ancillary analysis.*

Moreover, we did not exclude bilateral stroke patients. In fact, there were 21 patients with a lesion load of more than ten lesioned voxels in left- and right-hemispheric brain regions (please note that we applied a minimum cut-off to exclude incidental noise voxels). There were 38% female patients among these bilateral stroke patients, reflecting the same percentage of female patients in the entire sample.

*This information is added in **Figure caption 1 (p. 39)**:*

“Lesion atom 8 comprised several brain regions in the left hemisphere, however also featured the brainstem and, to a lesser degree, the right thalamus. This pattern likely arose since we did not exclude patients with bilateral stroke (n=21, 38% female).”

5) Were the NIHSS and MRI data collected in close temporal proximity? If not, this could also affect the results in the lesion-behavior correlation analysis.

*We thank the referee for this comment and are in agreement him/her that information on the exact time between MRI and NIHSS acquisition would have been ideal to conduct lesion-behavior analyses. However, as also outlined under **question 1**, to allow for a large-scale recruitment of stroke patients in clinical routine, inclusion criteria required a data acquisition during the first 48 hours after admission only and the exact time between image and stroke severity collection was not recorded. Since the majority of our data was not obtained on the first day of the hospital stay, drastic changes of the NIHSS score, e.g., due to revascularization therapy and spontaneous reperfusion, may have been less likely between the imaging session and NIHSS recording.*

*We furthermore aimed to increase insights on potential timing effects within the framework of our first ancillary analyses that considered more immediately acquired data only. As outlined under **question 1** more in detail, we could here recover female-specific effects for the left-hemispheric lesion atom 10.*

*As also pointed out under **question 1**, we now included this limitation in our **Discussion, Limitations and future directions section (p. 20)**:*

“Also, we did not have access to some measures, which could be of potential interest as covariate in our analysis, for a majority of patients; for example, delays in hospital arrival, the exact time between imaging and NIHSS acquisition, acute changes in stroke severity (e.g., due to spontaneous reperfusion, brain edema or seizures) or administered revascularization therapies. In this study, we aimed to address these factors indirectly by tailoring ancillary analyses to subgroups, e.g., only those patients whose data was acquired directly after admission. Future studies could recruit equal numbers of men and women, investigate these further covariates more explicitly”

6) The authors state DWI and FLAIR were co-registered for segmentation. Did these scans follow treatment? Were they performed in the hyper-acute setting for diagnosis? An MRI scan performed 2 hours after stroke onset cannot rely on a FLAIR sequence differently from DWI. The two sequences are more aligned at 48 hrs.

*We are grateful to the referee for having raised attention to our lesion segmentation and spatial normalization process. To clarify, we generated lesion masks using **only** the DWI sequences in both cohorts. The FLAIR images were primarily used to optimize the spatial normalization process and white matter lesion load estimation.*

We clarified these aspects in the revised version of our manuscript:

Methods, Stroke patient characteristics and imaging (p. 22):

“Ischemic DWI tissue lesions were manually outlined using semi-automated algorithms.”

Methods, Magnetic resonance imaging: Preprocessing (p. 23):

“The same transformation was applied to the DWI image as well as the corresponding **DWI-derived** binary lesion mask. **This preprocessing pipeline, especially featuring the co-registration of DWI and FLAIR images, was optimized to generate as reliable and accurate spatial normalizations for as many subjects as possible.**”

Table 1 (p. 44):

Normalized derived stroke lesion volume (ml)	DWI-	13.7 (29.9)	13.6 (31.8)	13.7 (28.7)	$p=0.98$
---	-------------	-------------	-------------	-------------	----------

Supplemental Materials, Validation cohort (p. 3):

“**DWI-defined** lesions were spatially normalized to MNI-space.”

*With respect to the question of whether scans were obtained before and after treatment, we would like to refer to **question 1**. We here outline ancillary analyses that aim to account for timing and treatment effects.*

7) The authors compare male vs. female patients, and found no statistical differences in lesion volume. However, it is not clear whether the frequency maps of lesions for male and females is similar. In other words, while the number of voxels affected in each parcel is not statistically different, do female and male lesions affect the same areas with the same frequency? This data and frequency maps for each subgroup need inclusion in the analysis (ideally displayed from 0 to maximum value).

We would like to thank the referee for his/her idea to additionally investigate potential sex differences in how often each brain region was affected. The exclusion of such frequency differences, which we present in the following, eventually contributes to increase the confidence to linking observed sex differences to sex-specific brain functions, instead of confounds due to sex- and sample-specific lesion patterns.

Results, Stroke sample characteristics (p. 8):

“Moreover, **neither the frequencies of how often each cortical and subcortical grey matter region or white matter tract was affected, nor the numbers of lesioned voxels within each region of interest** differed significantly between the sex categories, or between the left and right hemisphere (all **Fisher’s exact tests or two-sided t-tests: $p>0.05$** , Bonferroni-corrected for multiple comparisons, c.f., **Figure 1C, Supplementary Figure 1, Supplementary Tables 2 and 3**).”

Supplementary Materials, Supplementary Figure 1 (p. 2):

“Supplementary Figure 1. NIHSS distributions and frequency maps for the entire derivation cohort, as well as subgroups of only female and male patients.”

Further new figures and supplementary figures, displaying frequency maps for each of the samples investigated in derivation, validation and ancillary analyses, are displayed in the updated manuscript and supplementary materials documents. Please note that frequency maps start with at a minimum of zero, yet have varying maximum values that have been chosen to allow for a maximum differentiation of frequencies based on the colour map. Supplementary Tables 2 and 3, that now also contain information on the frequency tests, can be found in the updated supplementary materials.

8) This consideration is equally valid when the authors explore the effects of gender hormones on gender-specific differences, using a demographic categorization (age cutoff of 52 years). The larger effect in the older sample could be due to other factors such as a higher incidence of AF in older women. It is known that cardioembolic strokes affect more the ACA and PCA distribution (lesion atom 1 and 10).

*We thank the referee for identifying and highlighting the possibility of comparing region-wise frequencies in addition to lesion loads for the subgroups of male and female patients below and above the median age of menopause. We have now amended these comparisons without detecting any statistically significant differences (c.f., **Supplementary Figure 11** in the Supplementary Materials document).*

*Moreover, the results of our newly conducted ancillary analyses on cardioembolic vs. non-cardioembolic stroke genesis indicate that female-specific more pronounced lesion atom effects emerge independent of a cardioembolic stroke subtype (c.f., **question 3** for details and respective changes in the manuscript). Therefore, it seems unlikely that the constellation of present lesion atom effects for older female patients and missing ones for younger female patients stems from differences in stroke subtype frequencies.*

9) Since the main difference is in the left posterior (parietal) cortex, it is necessary to analyze the sub-scores of the NIHSS more related to the function carried out in that region. Language, dysarthria, and orientation scores should be more predictive than motor scores of the difference in lesion distribution.

We highly agree with the referee that it is an important next step to investigate sex-specific effects for more circumscribed brain functions, particularly since the spatial location of our most prominent female-specific effects suggest a link to functions other than motor functions.

We did not have access to NIHSS subitems in this large-scale study. However, given that sex-specific lesion pattern effects in stroke have rarely been examined, the main aim of our study was to derive and validate the mere existence of sex effects. Future studies are now warranted to assess the relationships between more specific neurological symptoms and lesion pattern.

*We outline these ideas for future research in **Discussion, Limitations and future directions (p. 19)**:*

“Furthermore, we here explored sex disparities in lesion patterns of *global* stroke severity, which allows for some broad clinical implications of our findings. **Naturally, our results may to a certain extent be dominated by effects due to motor symptoms, given their disproportionally high importance for the overall NIH Stroke Scale score.** To evaluate more specific brain functions at acute and chronic stages, future studies could **therefore** focus on sub-items of the NIHSS, **such as language impairments, dysarthria, disturbances in orientation, neglect or on specific cognitive behavioral tests, e.g., probing memory functions.** This would be a promising approach to trace back our most prominent sex-specific finding, the relevance of the left PCA-territory, to specific brain functions.”

10) The NIHSS scores' distribution should be displayed (in supplementary) separately for male, female, and female older and younger than 52.

*We thank the referee for the suggestion to add visualizations of NIHSS score distributions. Such visualizations can now be found for all of the subgroups considered in any of our analyses, amongst others the male and female patients older and younger than 52 (c.f., **Supplementary Figures 1, 5, 6, 8 and 11**).*

Exemplarily: Supplementary Materials, Supplementary Figure 8 (p. 9):

“Supplementary Figure 8. NIHSS distributions and frequency maps for subgroups of female and male patients with and without cardioembolic stroke.”

Overall, many substantial issues affect the main conclusion that differences in lesion distribution for males and females depend on brain substrates. This conclusion is by exclusion after all other factors mentioned above rule out. Since negative findings carry not as much weight as positive findings, it is necessary to rule out all issues above systematically before a conclusion about different brain substrates is acceptable with some confidence.

Minor:

11) The two datasets are unbalanced in terms of the ratio of male/female (more females). Why? This difference is surprising in a random sample.

We thank the referee for drawing attention to the unbalanced ratio of male and female patients in both the derivation and validation cohort (38% and 41% female patients, respectively). Such ratios actually resemble those ratios typically found in randomized clinical stroke trials (Carcel et al., 2019: Abstract: “Of 1700 stroke RCTs identified, 277 were published and eligible for analysis. Overall, these RCTs enrolled only 40% females, and in the past 10 years this percentage barely changed, peaking at 41% in 2008-2009 and 2012-2013.”)²⁵ and could be due to missing diagnoses of mild strokes in women (c.f., referee 3, question

2) or an under-recruitment of severely affected, elderly female stroke patients. The latter theory would be supported by a lower average age and stroke severity of female patients in our cohorts compared to unselected stroke registry cohorts (e.g., our derivation cohort's mean age in women: 67.7 years vs. the mean age in 365,331 German female stroke patients: 75.3 years and our derivation cohort's mean NIHSS in women: 5.6 vs. the mean NIHSS in 365,331 German female stroke patients: 7.1).²⁶

We now refer to this circumstance in the caption of **Table 1 (p. 44)**:

“The disproportionate representation of men and women may reflect an undersampling of female patients as frequently observed in randomized clinical stroke trials²⁵ and may largely stem from the non-inclusion of elderly and more severely affected female patients or more frequent missed diagnoses of mild stroke in women.”

We would like to point out that this undersampling was neither due to, nor amplified by excluding patients based on their lesion segmentation/spatial normalization quality, c.f., **Supplementary Materials, Derivation cohort (p. 1)**:

“We had access to 668 patients with manual lesion segmentations. Quality control of normalization results of structural images led to the exclusion of 55 out of these 668 patients. Included and excluded patients did not differ significantly with respect to age, sex and stroke severity (mean age(SD): 65.0(15.1) vs. 64.0(14.8), $p=0.66$, sex: 38% female vs. 35% female, $p=0.77$, mean NIHSS(SD): 5.04(6.0) vs. 6.1(6.0), $p=0.24$).”

Furthermore, we also observed female-specific lesion pattern effects in case of approximately balanced samples of men and women (c.f., ancillary analysis in patients with cardioembolic stroke featuring 47% female patients), which reinforced the notion that our findings were not artificially caused by the imbalance in men and women, c.f., **Results, Ancillary analysis (p. 13)**:

“Furthermore, all of these female-specific effects emerged for samples that were fairly even in their numbers of men and women (c.f., cardioembolic stroke: 47% female), or did not comprise any significant differences in age and stroke severity (c.f., non-cardioembolic stroke: two-sided t-tests: age: $p=0.45$, NIHSS: $p=0.44$), which additionally increased the confidence that sex differences did not artificially arise from these characteristics.”

Finally, we nonetheless included this aspect in our **Discussion, Limitations and future direction section (p. 20)**:

“However, our cohorts were slightly imbalanced with respect to the men:women ratio, which may not have faithfully captured the oftentimes reported equally high or higher stroke incidence in women.^{27,28} Also, we did not have access to some measures, which could be of potential interest as covariate in our analysis, for a majority of patients; for example, delays in hospital arrival, the exact time between imaging and NIHSS acquisition, acute changes in stroke severity (e.g., due to spontaneous reperfusion, brain edema or seizures) or administered revascularization therapies. In this study, we aimed to address these factors indirectly by tailoring ancillary analyses to subgroups, e.g., only those patients whose data was acquired directly after admission. Future studies could recruit equal numbers of men and women, investigate these further covariates more explicitly and additionally consider interaction effects of any of these variables with the patient's sex status.”

12) Was the stroke severity statistically higher in the female population based on the NIHSS scores? In both datasets? Please analyze.

We thank the referee for prompting to amend statistical comparisons of NIHSS scores for male and female patient subgroups.

As stated in the original manuscript version, male and female patients did not statistically significantly differ in their stroke severity based on a two-sided t-test with a level of significance of $p < 0.05$:

Table 1:

	All participants (n=555)	Women (n=208)	Men (n=347)	Statistical comparison of male and female patients
NIHSS	5.0(5.9) (median(iqr): 3 (6))	5.6(6.6) (median(iqr): 3 (6))	4.7(5.5) (median(iqr): 3 (5))	$p=0.09$

The same was found to be true for our validation cohort. In fact, the stroke severity between men and women was here also more similar in absolute terms.

Supplementary Table 1:

	All participants (n=503)	Women (n=204)	Men (n=299)	Statistical comparison of male and female patients
NIHSS	5.5(5.4) (median(iqr): 4(5))	5.8(5.6) (median(iqr): 4(6))	5.3(5.2) (median(iqr): 4(5))	$p=0.31$

Similarly, NIHSS-based stroke severity did not differ significantly for any of our men-women comparisons in ancillary analyses, but the comparison of men and women in the older age stratum.

Results, Ancillary analyses (p. 12):

*Data acquisition upon admission: “These criteria were fulfilled by 152 patients in total (men: 87, mean age(SD): 63.1(12.3), mean NIHSS(SD): 4.6(5.6); women: 65, mean age(SD): 69.6(14.5), mean NIHSS(SD): 5.5(6.5); two-sided t-tests: age: $p=0.003$, NIHSS: $p=0.45$, **Supplementary Figure 6).**”*

(Non-)Cardioembolic stroke: “203 patients with cardioembolic stroke: men: 108, mean age(SD): 66.6(14.3), mean NIHSS(SD): 6.3(5.4), women: 95, mean age(SD): 74.3(12.9), mean NIHSS(SD): 7.5(7.0); two-sided t-tests: age: $p < 0.001$, NIHSS: $p=0.17$; 677 patients with non-cardioembolic stroke: men: 419, mean age(SD): 62.8(13.3), mean NIHSS(SD): 4.7(5.3), women: 258, mean

age(SD): 63.7(16.4), mean NIHSS(SD): 5.1(5.7); two-sided t-tests: age: $p=0.45$, NIHSS: $p=0.44$, **Supplementary Figure 8**"

Below/Above 52 years of age: "below the age of 52 years (men: 113, mean age(SD): 43.1(8.5), mean NIHSS(SD): 5.4(6.1), women: 87, mean age(SD): 42.1(7.9), mean NIHSS(SD): 4.3(5.2); two-sided t-tests: age: $p=0.41$, NIHSS: $p=0.19$ [...] above the age of 52 years of age (**Supplementary Figure 12**, men: 533, mean age(SD): 68.4(9.5), mean NIHSS(SD): 4.9(5.2), women: 325, mean age(SD): 73.0(11.0), mean NIHSS(SD): 6.0(6.3); two-sided t-tests: age: $p<0.001$, NIHSS: $p=0.004$)."

13) The population size in the abstract is misleading. The authors should specify that the sample size was –500, and additional 500 subjects were part of a validation cohort.

We would like to thank the referee for this feedback and now clarify the precise number of subjects in each of the cohorts in the abstract, as well as everywhere in the manuscript.

Abstract (p. 4):

"This framework was tailored to carefully estimate possible sex differences in lesion patterns explaining acute stroke severity (NIHSS) in 555 patients (38% female). [...] These sex-sensitive lesion pattern effects were ~~discovered and subsequently~~ robustly validated in ~~two~~ a second large independent, multisite dataset (n=503, 41% female)."

Introduction (p. 7):

"For this purpose, we leveraged neuroimaging data originating from two large, independent hospital-based cohorts gathering data of 555 (derivation) and 503 (validation) acute ischemic stroke (AIS) patients in total."

Discussion (p. 14):

"In this study, we combined a novel probabilistic lesion-symptom mapping technique with empirical lesion data originating from two large independent cohorts of 555 (derivation) and 503 (validation) AIS patients to derive and validate sex-specific lesion patterns underlying NIHSS-based acute stroke severity."

14) A region-wise approach rather than a voxel-wise approach reduces the anatomical selectivity of the results. This issue needs discussion in the "Limitations and future directions" paragraph.

*We agree with the referee that this is an important limitation to bear in mind. We have thus extended our previous note in the **Discussion, Limitations and future directions section (p. 19)**:*

"Similarly, the spatial resolution of our Bayesian hierarchical approach stays within the realms of frequently arising, typical lesion patterns and back-transformed atlas brain regions. Consequently, our approach does not allow for a comparably high spatial resolution of lesion-symptom analyses relying on voxel-wise, instead of region-wise, data.^{2,4} Nonetheless, this reduction in lesion

dimensionality was necessary to render sex-aware analyses within our Bayesian hierarchical model framework feasible.”

Overall a provocative paper, but the main conclusions and most exciting conclusions are not robust yet.

Reviewer #3 (Remarks to the Author):

This study provides interesting data on the spatial distributions of stroke in men versus women, and how these differences contribute to stroke severity measured by the NIHSS. There are many strengths, including novel methodology (using Bayesian hierarchical linear regression), a large database, replication in a separate database. However, there are also some important weaknesses.

We would like to thank this referee for the positive evaluation of our presented manuscript.

1) First, many of the references are outdated, rendering some of the background “facts” incorrect. For example, the authors state, “For instance, women have a lower stroke incidence than men when younger, yet this initially low women-to-men incidence ratio is decisively reversed in the oldest age groups (>85 years).

However, according to the AHA website, women have a higher incidence of stroke than men. <https://www.stroke.org/en/about-stroke/stroke-risk-factors/women-have-a-higher-risk-of-stroke>

See also:

Madsen, T.E., Khoury, J., Alwell, K., Moomaw, C.J., Rademacher, E., Flaherty, M.L., Woo, D., Mackey, J., La Rosa, F.D.L.R., Martini, S. and Ferioli, S., 2017. Sex-specific stroke incidence over time in the Greater Cincinnati/Northern Kentucky Stroke Study. *Neurology*, 89(10), pp.990-996. These authors report, “Contrary to previous study periods, stroke incidence rates were similar by sex in 2010.”

See also:

Giroud, M., Delpont, B., Daubail, B., Blanc, C., Durier, J., Giroud, M. and Béjot, Y., 2017. Temporal trends in sex differences with regard to stroke incidence: the Dijon stroke registry (1987–2012). *Stroke*, 48(4), pp.846-849.

These authors conclude, “...more women than men experience an incident stroke each year because of a longer life expectancy.”

Furthermore, the age-adjusted incidence in women is also higher than men in some countries: Kim, J., Thayabaranathan, T., Donnan, G.A., Howard, G., Howard, V.J., Rothwell, P.M., Feigin, V., Norrving, B., Owolabi, M., Pandian, J. and Liu, L., 2020. Global stroke statistics 2019. *International Journal of Stroke*, p.1747493020909545.

These authors report, “The sex-specific age-adjusted incidence in China was greater in women than in men (309 vs. 280 per 100,000 population; online Supplementary Table 6).”

We would like to thank the referee for these further valuable insights and literature suggestions.

We have generally aimed to validate all our statements on previously determined sex differences by exhaustive literature search and alignment with central, recent reviews on sex differences in stroke presentation and care.^{29,30,31,32} As stated in these reviews, dependent on the outcome of interest, only few studies exist that have evaluated respective sex differences; sometimes these date back ~10 years (c.f.,

for example, Carcel et al.,: “A few studies have shown that nonconventional stroke symptoms are more prevalent in women than men (Gall et al., 2010; Di Carlo et al., 2003; Jerath et al., 2011).”³²

We decided to include the statement on a lower stroke incidence in younger women that is contrasted with a higher incidence in older women based on the Guidelines for the Prevention of Stroke in Women (2014): “Within most age strata, women have a lower IS incidence than men, and as such, the overall age-adjusted incidence of IS is lower for women than men^{4, 24, 26-31}; however, sex differences in IS incidence rates differ across the age strata. In the oldest age groups (generally >85 years of age), women tend to have higher^{12, 24, 27-30} or similar incidence of IS as men.^{4, 2}”²⁹

We do, however, agree with the referee that several sources, e.g., Madsen et al., 2017 and Kim et al., 2020, as stated by the referee, indicate an overall higher or similar incidence in women, while others, e.g., the referenced guidelines and Giroud et al., 2017 (as mentioned above by the referee), conclude on the opposite. In particular, Giroud and colleagues state in their abstract: “Despite lower rates, more women than men experience an incident stroke each year because of a longer life expectancy.” Realistically, the incidence of stroke may be influenced multi-factorially, e.g., by sex, yet also age, location and year of observation, which could explain the discrepant reports. In the Global Stroke Statistics 2019 (Kim et al., 2020) alone, the report on numbers in China (“The sex-specific age-adjusted incidence in China was greater in women than in men”) is followed by the one on numbers in India (“In contrast, the new age-adjusted stroke incidence rates were greater in men than in women in India”).

Eventually, we therefore decided upon referring to the higher absolute numbers of women experiencing a stroke each year, as this conclusion remains valid independent of potentially sex-specific rates of stroke, **Introduction (p. 6):**

“For instance, due to a longer life expectancy, more women than men experience a stroke each year.³³ Expected demographical changes, i.e., an aging population, will widen this gap further: in the US, projections suggest ~200.000 more disabled women after stroke than men by 2030.²⁹”

We furthermore point to the likely higher stroke incidence in women when discussing the limitations of our men:women ratio in the **Discussion (p. 20):**

“However, our cohorts were slightly imbalanced with respect to the men:women ratio, which may not have faithfully captured the oftentimes reported equally high or higher stroke incidence in women.^{27, 28}”

2) The higher incidence of stroke in women in most countries, together with the inclusion of only 37% women in the population studied raises the possibility of an important bias. That is, women with less symptomatic strokes may not have been included (because they were not identified as having stroke).

We are grateful to the referee for raising this point. While the low inclusion ratio of female patients is well in line with those ratios of randomized clinical stroke trials,²⁵ it is nonetheless important to consider as biasing factor.

We agree with the referee that missed diagnoses of mild stroke in women could be one reason of the lower numbers of female patients in our samples. As also outlined under **Minor 11 from referee 2**, another reason could be that severely affected, elderly female stroke patients have been under-recruited. This later

theory would be supported by a lower average age and stroke severity of female patients in our cohorts compared to unselected stroke registry cohorts (e.g., our derivation cohort's mean age in women: 67.7 years vs. the mean age in 365,331 German female stroke patients: 75.3 years and our derivation cohort's mean NIHSS in women: 5.6 vs. the mean NIHSS in 365,331 German female stroke patients: 7.1).²⁶

We addressed this topic of a potential sample bias severalfold: We, first of all, refer to the imbalance in men:women ratio and potential origins in the caption of **Table 1 (p. 44)**:

“The disproportionate representation of men and women may reflect an undersampling of female patients as frequently observed in randomized clinical stroke trials²⁵ and may largely stem from the non-inclusion of elderly and more severely affected female patients or more frequent missed diagnoses of mild stroke in women.”

We also ensured that the undersampling of women was neither due to, nor amplified by excluding patients based on their lesion segmentation/spatial normalization quality, c.f., **Supplementary Materials, Derivation cohort (p. 1)**:

“We had access to 668 patients with manual lesion segmentations. Quality control of normalization results of structural images led to the exclusion of 55 out of these 668 patients. Included and excluded patients did not differ significantly with respect to age, sex and stroke severity (mean age(SD): 65.0(15.1) vs. 64.0(14.8), $p=0.66$, sex: 38% female vs. 35% female, $p=0.77$, mean NIHSS(SD): 5.04(6.0) vs. 6.1(6.0), $p=0.24$).”

Furthermore, we also observed female-specific lesion pattern effects in case of approximately balanced samples of men and women that did not significantly differ in their stroke severity (c.f., ancillary analysis in patients with cardioembolic stroke featuring 47% female patients), which reinforced the notion that our findings were not artificially caused by the imbalance in men and women, c.f., **Results, Ancillary analysis (p. 13)**:

“Furthermore, all of these female-specific effects emerged for samples that were fairly even in their numbers of men and women (c.f., cardioembolic stroke: 47% female), or did not comprise any significant differences in age and stroke severity (c.f., non-cardioembolic stroke: two-sided t-tests: age: $p=0.45$, NIHSS: $p=0.44$), which additionally increased the confidence that sex differences did not artificially arise from these characteristics.”

Finally, we nonetheless included this aspect in our **Discussion, Limitations and future direction section (p. 20)**:

“However, our cohorts were slightly imbalanced with respect to the men:women ratio, which may not have faithfully captured the oftentimes reported equally high or higher stroke incidence in women.^{27,28} Also, we did not have access to some measures, which could be of potential interest as covariate in our analysis, for a majority of patients; for example, delays in hospital arrival, the exact time between imaging and NIHSS acquisition, acute changes in stroke severity (e.g., due to spontaneous reperfusion, brain edema or seizures) or administered revascularization therapies. In this study, we aimed to address these factors indirectly by tailoring ancillary analyses to subgroups, e.g., only those patients whose data was acquired directly after admission. Future studies could recruit equal numbers of men and women, investigate these further covariates

more explicitly and additionally consider interaction effects of any of these variables with the patient's sex status."

3) More importantly, the findings that both posterior (inferior division) MCA stroke and left PCA stroke were more common in women likely reflects the greater rate of atrial fibrillation in women. Atrial fibrillation is known to cause larger strokes as well as more frequent inferior division (compared to superior division) MCA strokes. This result accounts for the greater frequency of Wernicke's aphasia than Broca's aphasia in cardioembolic stroke. This result has been attributed to the greater accessibility of the inferior division than superior division to clots from the heart. Likewise, atrial fibrillation results in higher rate of posterior circulation (e.g. PCA) strokes than other common etiologies (such as carotid disease). Carotid stenosis (more common in men than women) is more likely to cause anterior (superior division) strokes.

Therefore, the results need to be re-analyzed by stroke etiology to determine if the findings can really be attributed to sex differences. The role of hormones is interesting, but purely speculative.

We appreciate this referee's suggestion to analyze the effect of stroke subtypes greatly. In combination with the second referee's suggestion (question 3), we have now added a completely novel ancillary analysis to disentangle the effects of cardioembolic versus non-cardioembolic strokes from sex-specific effects.

*The analysis is outlined in **Methods, Ancillary analyses (p. 27):***

"Conceivably, female-specific lesion effects could be due to a higher frequency of cardioembolic strokes and associated typical lesion patterns (e.g., more multifocal stroke lesions). In a second ancillary analysis, we therefore sought to disentangle stroke-subtype- and sex-specific effects and concurrently performed stroke-subtype- and sex-aware analyses, featuring two hierarchical levels with stroke subtype on the top level. Accordingly, patients were not only stratified into a group of men and women as in main analyses, yet initially assigned to a group of either cardioembolic or non-cardioembolic stroke genesis and then split into men and women (c.f., **Supplementary Materials** for a comprehensive display of model specifications). We then contrasted male- and female-specific lesion pattern effects within the cardioembolic and non-cardioembolic stroke subgroups. To allow for a reasonably high number of patients in each of these four subgroups, we here merged data of the derivation and validation cohorts (c.f., **Supplementary Materials** for further details)."

*Respective findings are described in **Results, Ancillary analyses (p. 12):***

"Women, especially if older, are known to more frequently experience atrial fibrillation and cardioembolic strokes.²⁴ The motivation of our second ancillary analysis was to explicitly consider the possibility that observed sex differences in explanatory lesion patterns merely stemmed from varying frequencies of cardioembolic strokes. As information on stroke subtype was available for subgroups of each cohort only, we here pooled data from the derivation and validation cohort to rest analyses on a sample as large as possible. Approximately one fourth of the 880 patients in total experienced a cardioembolic stroke (203 patients with cardioembolic stroke: men: 108, mean age(SD): 66.6(14.3), mean NIHSS(SD): 6.3(5.4), women: 95, mean age(SD): 74.3(12.9), mean NIHSS(SD): 7.5(7.0); two-sided t-tests: age: $p < 0.001$, NIHSS: $p = 0.17$; 677 patients with non-cardioembolic stroke: men: 419, mean age(SD): 62.8(13.3), mean NIHSS(SD): 4.7(5.3), women:

258, mean age(SD): 63.7(16.4), mean NIHSS(SD): 5.1(5.7); two-sided t-tests: age: $p=0.45$, NIHSS: $p=0.44$, **Supplementary Figure 8**). As expected, women featured a higher frequency of cardioembolic strokes (Fisher's exact test: $p=0.03$). We detected substantial female-specific *lesion atom* effects independent of whether comparing men and women with cardioembolic strokes, or men and women with non-cardioembolic stroke genesis: In case of cardioembolic stroke, difference distributions of *lesion atoms* 4, 6 and 8 suggested a higher explanatory relevance exclusively in women (**Supplementary Figure 9**), while *lesion atoms* 1, 2, 9 and 10 had a substantially higher relevance in women in case of non-cardioembolic stroke (**Supplementary Figure 10**). Since we extracted female-specific effects in strata of both only cardioembolic and non-cardioembolic stroke patients, these results rendered the interpretation of lesion pattern effect differences due to varying frequencies of cardioembolic stroke unlikely. Furthermore, all of these female-specific effects emerged for samples that were fairly even in their numbers of men and women (c.f., cardioembolic stroke: 108 men and 95 women), or did not comprise any significant differences in age and stroke severity (c.f., non-cardioembolic stroke: two-sided t-tests: age: $p=0.45$, NIHSS: $p=0.44$), which additionally increased the confidence that sex differences did not artificially arise from these differences."

*Further details can be found in **Supplementary Materials, Ancillary analyses: Stratifying for cardioembolic stroke genesis (p. 8)**:*

"Data of the derivation and validation cohorts were merged in ancillary analyses focused on the stroke subtype and age of participants. More specifically, we adopted the NMF-transformation as learned exclusively based on the derivation cohort and applied the same transformation to the validation cohort. To harmonize data of both cohorts further, we normalized NMF-transformed lesion data before merging and inserting it into the linear regression model. We refrained from re-computing the NMF-transformation based on the merged data and chose to rely on the derivations cohort's lesion embedding instead to increase the ease of interpretation: In this way, any results could immediately be compared to the derivation cohort's results. This choice was furthermore supported by our analyses demonstrating the independence of the concrete lesion embedding (c.f., **Results: Validation analyses**). Lastly, we added cohort membership as a covariate to the model to further account for differences between cohorts."

Newly added supplementary figures, as well as the full model specifications can be found in the revised supplementary materials document.

In view of these results, we thus feel confident to conclude that female-specific lesion atom effects arose independent from cardioembolic or non-cardioembolic stroke subtypes. Indeed, the higher relevance of lesion atom 10 in women, the most reliably detectable effect, emerged in the subgroup of non-cardioembolic stroke patients and was thus not depending on a cardioembolic stroke-related stroke lesion pattern.

4) The finding of stronger sex-specific effects in older patients and no sex-specific effects in younger patients might again reflect differences in etiology of stroke (more atrial fibrillation in women) rather than sex differences.

We thank the referee for pointing out further confounds due to stroke subtype in our age-stratified analyses.

Given that the previous comment 3 motivated new ancillary analyses, that we decided to conduct based on a merged dataset to increase each subgroup's number of contributing patients, we additionally renewed our age-stratified analyses: We here now decided to also rely on the merged dataset of derivation and validation cohort. This decision resulted in substantially increased numbers of patients in all subgroups. Importantly, this increase rendered the men – women comparison in younger (<52 years) patients more reliable in particular.

Since we could ascertain several female-specific lesion atom effects in case of both cardioembolic, as well as non-cardioembolic stroke subtype, we would also expect to find female-specific effects in both age strata, independent of lower or higher rates of atrial fibrillation.

However, what we find is that several female-specific lesion atom effects exist in patients above the median age of menopause, yet none remain observable in patients below the median age of menopause. Quite conversely, we actually observed a male-specific lesion atom effect in patients below the median age of menopause.

All in all, we feel that the joint consideration of our ancillary analyses speaks against the interpretation that differences in the younger and older age stratum were merely due to differences in stroke subtype. Nevertheless, this analysis certainly has limitations, especially relating to a more granular evaluation of the hormonal status, that could be addressed in future research.

*We outline these considerations more in detail in our updated **Discussion, Limitations and future directions (p. 18, c.f., also referee 4, question 1):***

“In this study, we did not have access to information on the hormonal status in women. Instead, we here assumed the same median age of menopause for all female patients. Employing the actual age at menopause, as well as increasing the number of younger patients in general would have allowed for stronger conclusions on the potential organizational or activational character of findings. Besides, the availability of the exact level of estrogens would be most desirable, as previous studies indicate that sex-specific functional cerebral asymmetries may even vary during the menstrual cycle.³⁴ Moreover, the measurement of exact hormonal levels could inform about the effects of hormone therapy (e.g., hormone replacement therapy after menopause, oral contraceptives prior to menopause) and surgical interventions, such as hysterectomies, oophorectomies and ovaries remaining after hysterectomy. Therefore, future studies could attempt to gather data on hormonal status and also to recruit more balanced numbers of men and women of pre- and postmenopausal status. Additionally, it might be promising to test links between implicated brain regions and sex-specific genetic underpinnings, as was recently introduced for healthy population samples.^{35,36}”

*In the following, we would like to add the updated age-stratified analysis: **Methods, Ancillary analyses (p. 28):***

“Lastly, we aimed to explore whether possible sex differences were more likely to be of organizational or activational hormonal nature. Since we did not have access to the precise hormonal status in women, we stratified our sample based on an age cutoff of 52 years, according to the median age at menopause.³⁷ As in the ancillary analysis focused on stroke subtype, we inserted an additional hierarchical level and build two groups of patients above (\geq) 52 and below (<52) years of age within which we compared lesion pattern effects of men and women. We once

again merged derivation and validation cohort data to maximize the number of available patients, especially in the subgroup of younger stroke patients.”

*Findings are described in **Results, Ancillary analyses (p. 13):***

“Finally, we aimed to explore the **potential** effects of sex hormones, such as estrogen, which are known to be markedly affected by menopause.³⁸ We stratified the entire group of patients according to their sex and an age cutoff of 52 years, the median age at menopause.³⁷ **All of the female-specific *lesion atom* effects, as observed in the main analysis, disappeared in the analysis of all men and women below the age of 52 years (men: 113, mean age(SD): 43.1(8.5), mean NIHSS(SD): 5.4(6.1), women: 87, mean age(SD): 42.1(7.9), mean NIHSS(SD): 4.3(5.2); two-sided t-tests: age: $p=0.41$, NIHSS: $p=0.19$; **Supplementary Figure 11**). What is more, *lesion atom 7* was now assigned a higher relevance in male patients (**Supplementary Figure 12**). In contrast, we observed female-specific higher relevances in three *lesion atoms* (*lesion atoms 1, 9 and 10*), when comparing men and women in the subgroup of patients above the age of 52 years of age (**Supplementary Figure 13**, men: 533, mean age(SD): 68.4(9.5), mean NIHSS(SD): 4.9(5.2), women: 325, mean age(SD): 73.0(11.0), mean NIHSS(SD): 6.0(6.3); two-sided t-tests: age: $p<0.001$, NIHSS: $p=0.004$). Overall, female-specific effects were thus noticeably more pronounced with advanced age. Given that we ascertained female-specific effects for both cardioembolic, as well as non-cardioembolic stroke subtypes in the previous ancillary analysis, the observations for older female patients are unlikely due to an increased frequency of cardioembolic stroke and linked lesion patterns. Further, this observable *sex-age* interaction might also explain why we discovered more extensive female-specific effects in the derivation cohort, featuring significantly older female patients, than in the validation cohort that was characterized by a non-significant age difference between men and women.”**

And newly created Supplementary Figures 11-13 can be found in the Supplementary Materials document.

Reviewer #4 (Remarks to the Author):

The authors have performed detailed analyses of sex-specific lesions in men and women with acute ischemic stroke. They claim a novel interpretation for why women may have more severe strokes (based on NIHSS) compared to men, and this relates to specific regions of the brain that differ by sex. The authors also performed analyses by stratifying by age, choosing 52 yrs as the strata due to the age at menopause.

Overall, these results are very fascinating since this detailed mapping of lesions based on severity scores clearly demonstrate sex differences. I have concerns that relate to the hormonal status analyses.

We would like to thank the referee for the positive comments and constructive criticism.

1) Some women use hormone therapy past age 52, so the limitations need to include this fact. Also, many women under age 52 could have been using oral contraceptives. There are also nuances related to age at hysterectomy/oophorectomy and ovaries remaining after hysterectomy that need to be acknowledged.

We appreciate these thoughtful suggestions on further sources of hormonal changes greatly. As suggested by this referee, we now outline these considerations in the **Discussion, Limitations and future directions section (p. 18)**:

“Besides, the availability of the exact level of estrogens would be most desirable, as previous studies indicate that sex-specific functional cerebral asymmetries may even vary during the menstrual cycle.³⁴ Moreover, the measurement of exact hormonal levels could inform about the effects of hormonotherapy (e.g., hormone replacement therapy after menopause, oral contraceptives prior to menopause) and surgical interventions, such as hysterectomies, oophorectomies and ovaries remaining after hysterectomy.”

2) In addition, the women were older than men at the time of the stroke, and it is not clear how this was taken into account in the overall analyses.

We would like to thank this referee for pointing out the importance of age in our analyses and we apologize for previously insufficiently presenting how we incorporated this information in our analyses.

Especially since stroke populations are innately also populations of older patients, we ensured to account for age effects in several ways (c.f., also **referee 1, question 7**):

We directly controlled for age by adding it as covariate in our Bayesian model: **Methods, Explaining inter-individual differences in acute stroke outcomes (p. 24)**:

“In view of women’s overall higher age, we aimed to adjust for age-specific effects particularly comprehensively by not only taking into account the basic age, yet also its squared value.”

We furthermore aimed to indirectly address age effects, such as cerebral atrophies, by ensuring a high quality of each included, spatially normalized lesion segmentation: **Methods, Magnetic resonance imaging: Preprocessing (p. 23)**:

“This preprocessing pipeline, especially featuring the co-registration of DWI and FLAIR images, was optimized to generate as reliable and accurate spatial normalizations for as many subjects as possible. The quality of normalized lesion masks was carefully inspected by two experienced raters (A.K.B, M.B).”

Notably, our spatial normalization step compensated for varying brain sizes. Next, we also accounted for further clinical covariates, such as the white matter hyperintensity lesion load, which may be regarded as proxies of chronic brain health and biological brain age. Moreover, we would like to add that we also observed female-specific effects in samples of men and women that did not differ in age (e.g., the validation cohort and the non-cardioembolic stroke stratum) and should thus be affected by similar age effects.

After all, the more advanced age of women in our derivation cohort compared to the age of women in our validation cohort combined with the findings of our ancillary analyses (suggesting more pronounced female-specific effects for patients above 52 years of age), could however explain why we generally detected more substantially altered lesion atoms in our derivation cohort compared to our validation cohort. We express these considerations in **Results, Ancillary analyses (p. 13)**:

“Further, this observable *sex-age* interaction might also explain why we discovered more extensive female-specific effects in the derivation cohort, featuring significantly older female patients, than in the validation cohort that was characterized by a non-significant age difference between men and women.”

3) The low prevalence of hypertension was quite surprising--only 28%?

We agree with the referee that the prevalence of hypertension was comparatively low in our derivation cohort, given that unselected stroke registry samples frequently record a prevalence of ~70% (e.g., 74% in 259,845 Get With the Guidelines-Stroke Program patients, Smith et al., 2010).²¹ However, our validation cohort was characterized by a substantially higher prevalence of hypertension, i.e., 64% overall. We now present this value in a newly added supplementary table 1 comprising detailed information on sociodemographic and clinical characteristics of the validation cohort.

Importantly, main findings, i.e., substantially diverging lesion atom effects in men and women, remained the same independent of inclusion or exclusion of covariates on comorbidities in case of both our derivation, as well as validation cohort. These observations suggest that varying prevalences on comorbidities do not markedly alter lesion atom effects.

We now present these considerations in **Supplementary Materials, Supplementary Table 1 (p. 3)**:

“**Supplementary Table 1. Patient characteristics of the validation cohort.** Mean (SD) unless otherwise noted. The groups of male and female patients were compared via two-sample t-tests or Fisher’s exact test as appropriate. Asterisks indicate significant differences between men and women. Substantially more patients of the validation cohort had a history of hypertension, atrial fibrillation and coronary artery disease in comparison to the derivation cohort; potentially reflecting veridical sampling differences or differences in data acquisition. Importantly, sex-specific *lesion atom* effects of both derivation and validation cohorts remained the same when excluding comorbidities from the list of covariates (data not shown).”

	All participants (n=503)	Women (n=204)	Men (n=299)	Statistical comparison of male and female patients
Age	65.0(14.6)	65.3(16.3)	64.8(13.2)	$p=0.70$
Sex	59% male,41% female	-	-	
NIHSS	5.5(5.4) (median(iqr): 4(5))	5.8(5.6) (median(iqr): 4(6))	5.3(5.2) (median(iqr): 4(5))	$p=0.31$
Normalized DWI-derived stroke lesion volume (ml)	21.0(40.5) (median(iqr): 3.9(19.8))	22.9(38.7) (median(iqr): 4.0(31.3))	19.7(41.7) (median(iqr): 3.7(15.0))	$p=0.38$

White matter hyperintensity lesion volume (ml)	11.13(13.6) (median(iqr): 5.8(12.3))	9.6(11.6) (median(iqr): 4.9(13.2))	12.2(14.8) (median(iqr): 6.2(12.4))	$p=0.04^*$
Hypertension	63.6%	67.2%	61,2%	$p=0.19$
Diabetes mellitus type 2	22.7%	20.6%	24.1%	$p=0.39$
Atrial fibrillation	19.3%	24.0%	16.1%	$p=0.03^*$
Coronary artery disease	16.9%	11.8%	20.4%	$p=0.01^*$

4) Understanding the impact of this imaging analysis would be quite powerful if there were functional clinical outcomes at 30 or 90 days. Perhaps this is the follow-up plan in these cohorts, but it would be a nice validation of cognitive, fatigue, or other quality of life outcomes that could be compared between men and women.

We are in high agreement with the referee and thank him/her for this suggestion. The aims of our current study were to introduce instrumental new methodology to investigate lesion-pattern-sex interactions in the first place and examine whether sex differences existed in the acute phase. Given that we here display evidence of acute female-specific lesion pattern effects, determining the lasting effect of sex-specific differences is one of the most vital next steps. It is, however, beyond the scope of the current project.

*We motivate these future research ideas in **Discussion, Limitations and future directions (p. 19)**:*

“Furthermore, we here explored sex disparities in lesion patterns of *global* stroke severity, which allows for some broad clinical implications of our findings. Naturally, our results may to a certain extent be dominated by effects due to motor symptoms, given their disproportionately high importance for the overall NIH Stroke Scale score. To evaluate more specific brain functions at acute and chronic stages, future studies could therefore focus on sub-items of the NIHSS, such as language impairments, dysarthria, disturbances in orientation, neglect or on specific cognitive behavioral tests, e.g., probing memory functions. This would be a promising approach to trace back our most prominent sex-specific finding, the relevance of the left PCA-territory, to specific brain functions. Incorporating outcomes from more chronic time points would furthermore allow for more definite conclusions on long-term effects and, for example, their socioeconomic meaning. Above all, it may be especially fruitful to examine whether sex differences observed here generalize to cerebral reorganization processes during the recovery phase post-stroke. In the positive case, generated results could fuel sex-specific personalized clinical rehabilitation endeavors.²⁰”

References

1. Lee, D. D. & Seung, H. S. Learning the parts of objects by non-negative matrix factorization. *Nature* **401**, 788 (1999).
2. Bates, E. *et al.* Voxel-based lesion–symptom mapping. *Nature neuroscience* **6**, 448 (2003).

3. Vaidya, A. R., Pujara, M. S., Petrides, M., Murray, E. A. & Fellows, L. K. Lesion Studies in Contemporary Neuroscience. *Trends in cognitive sciences* (2019).
4. Smith, D. V., Clithero, J. A., Rorden, C. & Karnath, H.-O. Decoding the anatomical network of spatial attention. *Proceedings of the National Academy of Sciences* **110**, 1518–1523 (2013).
5. Zhang, Y., Kimberg, D. Y., Coslett, H. B., Schwartz, M. F. & Wang, Z. Multivariate lesion-symptom mapping using support vector regression. *Human brain mapping* **35**, 5861–5876 (2014).
6. Yourganov, G., Fridriksson, J., Rorden, C., Gleichgerrcht, E. & Bonilha, L. Multivariate Connectome-Based Symptom Mapping in Post-Stroke Patients: Networks Supporting Language and Speech. *The Journal of Neuroscience* **36**, 6668–6679 (2016).
7. Pustina, D. *et al.* Enhanced estimations of post-stroke aphasia severity using stacked multimodal predictions: Enhanced Predictions of Aphasia Severity. *Human Brain Mapping* **38**, 5603–5615 (2017).
8. Toba, M. N. *et al.* Game theoretical mapping of causal interactions underlying visuo-spatial attention in the human brain based on stroke lesions. *Human Brain Mapping* **38**, 3454–3471 (2017).
9. Sperber, C. Rethinking causality and data complexity in brain lesion-behaviour inference and its implications for lesion-behaviour modelling. *Cortex* **126**, 49–62 (2020).
10. Mah, Y.-H., Husain, M., Rees, G. & Nachev, P. Human brain lesion-deficit inference remapped. *Brain* **137**, 2522–2531 (2014).
11. Van Den Heuvel, M. P. & Sporns, O. Rich-club organization of the human connectome. *Journal of Neuroscience* **31**, 15775–15786 (2011).
12. Schirmer, M. D. *et al.* Rich-Club Organization: An Important Determinant of Functional Outcome After Acute Ischemic Stroke. *Front. Neurol.* **10**, (2019).
13. Godefroy, O. *et al.* Brain-behaviour relationships. Some models and related statistical procedures for the study of brain-damaged patients. *Brain: a journal of neurology* **121**, 1545–1556 (1998).

14. Toba, M. N. *et al.* *Revisiting 'brain modes' in a new computational era: approaches for the characterization of brain-behavioural associations.* (Oxford University Press, 2020).
15. Tuladhar, A. M. *et al.* Relationship between white matter hyperintensities, cortical thickness, and cognition. *Stroke* **46**, 425–432 (2015).
16. Desikan, R. S. *et al.* An automated labeling system for subdividing the human cerebral cortex on MRI scans into gyral based regions of interest. *Neuroimage* **31**, 968–980 (2006).
17. Mori, S., Wakana, S., Van Zijl, P. C. & Nagae-Poetscher, L. M. *MRI atlas of human white matter.* (Elsevier, 2005).
18. Bonkhoff, A. K. *et al.* Generative lesion pattern decomposition of cognitive impairment after stroke. *bioRxiv* (2020).
19. Albers, G. W. *et al.* Thrombectomy for Stroke at 6 to 16 Hours with Selection by Perfusion Imaging. *New England Journal of Medicine* **378**, 708–718 (2018).
20. Nachev, P., Rees, G. & Frackowiak, R. Lost in translation. *F1000Research* **7**, 620 (2018).
21. Smith, E. E. *et al.* Risk Score for In-Hospital Ischemic Stroke Mortality Derived and Validated Within the Get With The Guidelines–Stroke Program. *Circulation* **122**, 1496–1504 (2010).
22. Gatttringer, T. *et al.* Predicting early mortality of acute ischemic stroke: score-based approach. *Stroke* **50**, 349–356 (2019).
23. Sharobeam, A. *et al.* Patterns of Infarction on MRI in Patients With Acute Ischemic Stroke and Cardio-Embolic: A Systematic Review and Meta-Analysis. *Frontiers in neurology* **11**, 1699 (2020).
24. Friberg, L., Benson, L., Rosenqvist, M. & Lip, G. Y. H. Assessment of female sex as a risk factor in atrial fibrillation in Sweden: nationwide retrospective cohort study. *BMJ* **344**, e3522–e3522 (2012).
25. Carcel, C. *et al.* Trends in recruitment of women and reporting of sex differences in large-scale published randomized controlled trials in stroke. *International Journal of Stroke* **14**, 931–938 (2019).

26. Bonkhoff A.K., Karch A., Weber R., Wellmann J., & Berger K. The female stroke - explaining sex differences in acute treatment and early outcomes of acute ischemic stroke patients. *Stroke (in press)* (2021).
27. Madsen, T. E. Sex-specific stroke incidence over time in the Greater Cincinnati/Northern Kentucky Stroke Study. *intracerebral hemorrhage* 9 (2017).
28. Kim, J. *et al.* Global stroke statistics 2019. *International Journal of Stroke* 1747493020909545 (2020).
29. Bushnell, C. *et al.* Guidelines for the Prevention of Stroke in Women: A Statement for Healthcare Professionals From the American Heart Association/American Stroke Association. *Stroke* **45**, 1545–1588 (2014).
30. Bushnell, C. *et al.* Sex differences in the evaluation and treatment of acute ischaemic stroke. *The Lancet Neurology* **17**, 641–650 (2018).
31. Bushnell, C. D. *et al.* Sex differences in stroke: Challenges and opportunities. *Journal of Cerebral Blood Flow & Metabolism* **38**, 2179–2191 (2018).
32. Carcel, C., Woodward, M., Wang, X., Bushnell, C. & Sandset, E. C. Sex matters in stroke: A review of recent evidence on the differences between women and men. *Frontiers in Neuroendocrinology* **59**, 100870 (2020).
33. Giroud, M. *et al.* Temporal Trends in Sex Differences With Regard to Stroke Incidence: The Dijon Stroke Registry (1987–2012). *Stroke* **48**, 846–849 (2017).
34. Hausmann, M. Why sex hormones matter for neuroscience: A very short review on sex, sex hormones, and functional brain asymmetries. *Journal of Neuroscience Research* **95**, 40–49 (2017).
35. Liu, S., Seidlitz, J., Blumenthal, J. D., Clasen, L. S. & Raznahan, A. Integrative structural, functional, and transcriptomic analyses of sex-biased brain organization in humans. *Proceedings of the National Academy of Sciences* 201919091 (2020) doi:10.1073/pnas.1919091117.
36. Smith, S. M. *et al.* Enhanced Brain Imaging Genetics in UK Biobank. *bioRxiv* (2020).

37. McKinlay, S. M., Brambilla, D. J. & Posner, J. G. The normal menopause transition. *Maturitas* **14**, 103–115 (1992).
38. Koellhoffer, E. C. & McCullough, L. D. The Effects of Estrogen in Ischemic Stroke. *Translational Stroke Research* **4**, 390–401 (2013).

REVIEWER COMMENTS

Reviewer #1 (Remarks to the Author):

I thank the authors for having carefully considered my comments. The paper is now improved with more methodological details and integration of key previous literature.

Reviewer #3 (Remarks to the Author):

The authors have very thoughtfully responded to all of the previous criticisms

Reviewer #4 (Remarks to the Author):

The authors present revised analyses, as well as extensive comments to each reviewer concern. Although limitations to the overall conclusions remain, such as hormonal status, and timing of imaging in relation to initial NIHSS and acute stroke interventions, the findings in this study contribute to current knowledge regarding sex differences in stroke. I have no further requests for revisions.

Reviewer #5 (Remarks to the Author):

Thank you for inviting me to review this manuscript.

I am enthusiastic about this study for its sizable independent datasets and novel analysis. However, as a stroke researcher, I think it is a shame that the authors missed an excellent opportunity to take these data to the next level and include more relevant clinical information. For example, acute treatment, the delay in arrival time to the hospital, and the interaction between NIHSS assessment and treatment (e.g. NIHSS severity on admission and after acute treatment can be hugely different) are critical measures to take into consideration. In combination, these omissions make statements like 'Women tend to have more severe strokes than man.' (first sentence of the conclusion) a little tricky as they may have simply arrived later at the hospital and hence not received the same acute treatments, ultimately leading to more severe manifestations of symptoms.

As it stands, this manuscript is a valid contribution to the community. Still, the manuscript's quality and clinical impact would benefit a lot if the authors could add and analyse the missing data to improve their predictive model's clinical relevance.

Overall, I believe the authors have addressed the reviewers comments but with the caveat that a lot of the data is not accessible.

We would like to thank you for the considerate evaluation of our initial revision and are grateful for new thoughtful feedback on our updated manuscript, "Sex-specific lesion topographies explain outcomes after acute ischemic stroke".

In this revision, we took great care to more accurately outline how we addressed potential effects of further clinical characteristics of interest, such as delays in hospital arrival and acute treatment effects on stroke severity. In particular, we now highlight that all models are adjusted for global, lesion pattern-independent sex effects (e.g., potential negative effects due to female-specific more delayed hospital arrival) at multiple instances in the manuscript.

We address all concerns raised by the fifth referee in a point-to-point fashion. Reviewer queries are in bold, while our responses are in non-bold and italics. New material is in red and bold in the revised manuscript and furthermore reproduced in this revision letter. Once again, we believe that the revised manuscript is substantially stronger as a result of the helpful comments.

Reviewer #1 (Remarks to the Author):

I thank the authors for having carefully considered my comments. The paper is now improved with more methodological details and integration of key previous literature.

Reviewer #3 (Remarks to the Author):

The authors have very thoughtfully responded to all of the previous criticisms.

Reviewer #4 (Remarks to the Author):

The authors present revised analyses, as well as extensive comments to each reviewer concern. Although limitations to the overall conclusions remain, such as hormonal status, and timing of imaging in relation to initial NIHSS and acute stroke interventions, the findings in this study contribute to current knowledge regarding sex differences in stroke. I have no further requests for revisions.

Thank you, we appreciate these referees' positive and promising evaluations of our revised manuscript greatly.

Reviewer #5 (Remarks to the Author):

Thank you for inviting me to review this manuscript.

I am enthusiastic about this study for its sizable independent datasets and novel analysis. However, as a stroke researcher, I think it is a shame that the authors missed an excellent opportunity to take these data to the next level and include more relevant clinical information. For example, acute treatment, the delay in arrival time to the hospital, and the interaction between NIHSS assessment and treatment (e.g. NIHSS severity on admission and after acute treatment can be hugely different) are critical measures to take into consideration.

We thank the referee for expressing enthusiasm for our work. Furthermore, we are in high agreement with the referee that the explicit consideration of information on further clinical aspects, such as delayed

hospital arrival, acute treatments and sex-specific treatment effects on stroke severity, would have been ultimately desirable. However, this information was not systematically available to us; neither in the derivation, nor in the validation cohort.

Therefore, we chose indirect strategies to address these aspects as comprehensively as possible:

For example, we conducted ancillary analyses focusing on exclusively those patients with data acquisition upon admission (and not within the first 48h after admission, as ensured for the entire cohort). By these means, we minimized any potential biases due to stroke-severity altering treatment effects. These analyses confirmed our main finding: Left-sided lesions in the posterior circulation led to a higher stroke severity specifically in women (c.f. **methods, p. 26 and results, p. 10** for more details on these analyses).

Motivated by the referee's comment, we have now extended our limitation section with respect to treatment effects on stroke severity and have also optimized our reference to the just mentioned ancillary analyses, their motivation and results in the updated **Discussion (p. 19)**:

“Also, we did not have access to some measures [...] for a majority of patients; for example, [...] administered revascularization therapies and their interactions with stroke severity (e.g., NIHSS scores upon admission and after acute treatment may differ substantially). In this study, we aimed to address these factors indirectly by tailoring ancillary analyses to subgroups, e.g., only those patients whose data was acquired directly after admission. Their data acquisition thus preceded the onsets of any potential acute treatment effects with high probability – female-specific effects relating to left posterior circulation lesions were nonetheless reliably observable.”

In addition, while several of the current authors have worked with various (of the overall few) large stroke imaging datasets,^{1,2} we are not aware of any larger dataset that comprises readily available information on both spatially normalized lesions as well as acute treatments (and particularly not the timing in relation to image acquisition!). Thus, the acquisition of such a dataset may be a pertinent and very relevant future aim and we agree with the referee that it would take related research projects to the next level.

We now feature the need for datasets with extended clinical information and associated novel possibilities in the revised **Discussion (p. 19)**:

“Future large-scale studies are warranted to recruit equal numbers of men and women and systematically record further clinical aspects, such as delays in hospital arrival and acute treatments, to facilitate more comprehensive and explicit investigations. In particular, these studies may be enriched for patients receiving acute treatments to maximize the power of uncovering sex- and treatment-specific lesion atom-effects. By these means, it may eventually be possible to quantify the clinical importance of sex-specific lesion patterns by taking into account their modifying effects on acute recanalization treatments.”

In fact, demonstrating that the awareness of sex-specific lesion pattern contributions to acute stroke severity, as uncovered here, can optimize acute stroke management, particularly the decisions on acute treatments, may eventually be one of the clinically most powerful and promising aspects of our study. A study published in *JAMA Neurology* just a month ago indicated that thrombectomy of primary distal posterior cerebral artery occlusions may be safe and show beneficial effects on stroke severity.³ These

exciting results may now spur and facilitate further investigations of our sex-specific lesion pattern effects in clinical practice even more!

*Accordingly, we have optimized the respective text in the **Discussion (p. 17)**:*

*“Furthermore, it may be important to take into account this female-specific salvaging effect for lesions in the posterior circulation territory of the left hemisphere. **Thrombectomy for Primary Distal Posterior Cerebral Artery Occlusion Stroke was recently found to be a safe and potentially beneficial treatment option, as treated patients showed a pronounced decrease in stroke severity until discharge.³ In view of these promising new results, future thrombectomy studies could evaluate whether female patients benefited even more substantially from these reperfusion therapies of more distal PCA-occlusions and whether there should be a higher tendency to offering this kind of treatment.**”*

In combination, these omissions make statements like ‘Women tend to have more severe strokes than men.’ (first sentence of the conclusion) a little tricky as they may have simply arrived later at the hospital and hence not received the same acute treatments, ultimately leading to more severe manifestations of symptoms.

We would like to thank the referee for prompting further clarifications on how we addressed and adjusted for overall sex differences. Importantly, our employed adjustment can take into account even latent factors, i.e. those factors that we did not explicitly measure in our cohort, such as varying hospital arrival times.

In addition to modeling male- and female-specific lesion pattern effects (i.e., interaction effects), we also included sex itself as a covariate (i.e., main effect). Hence, our lesion pattern-dependent results are corrected for any effects concerning all women or all men independent of specific lesion patterns.

To increase the visibility of these very important methodological details, we have added and extended respective information in several parts of the manuscript:

Results (p. 8):

*“Notably, we included sex as a variable in the model to differentiate between sex differences in stroke severity that were *dependent* and *independent* of lesion patterns. If, for example, stroke severity was generally higher in women, without any link to the actual lesion pattern, **conceivably due to a longer delay between symptom onset and hospital admission and decreased likelihood of acute treatment administration**, this effect would be represented by the Bayesian posterior distribution of this sex variable, but not in the sex-specific *lesion atom* Bayesian posterior distribution. [...] **Lastly, it is important to note that this adjustment for global sex differences was independent of the exact knowledge or measurements of their causes, i.e., we did not need to include any information on the delay in hospital admission explicitly.**”*

Discussion (p. 18):

“We furthermore adjusted all analyses for global, *lesion atom-independent* sex differences (e.g., due to a potentially longer time between symptom onset and hospital admission in women).”

Methods (p. 23):

“Importantly, we also included sex as a non-hierarchical, lower-level variable in the model to capture sex differences in stroke severity that were *independent* of lesion patterns. Women may, for example, also present with more severe strokes on average. **They may be of a more advanced age and have a greater pre-stroke level of disability when experiencing the stroke, leading to a higher stroke severity, yet decreased likelihood of acute recanalization treatment.** Such sex-specific, but lesion pattern-independent differences in stroke severity could then be modeled by the sex variable. It would, however, not have an influence on *lesion pattern-sex* interaction effects.”

Figure 5 (p. 43):

“All analyses were corrected for overall sex effects, i.e., *lesion atom-independent* effects (e.g., due to potential female-specific delayed hospital arrival). Of note, this correction was independent of the explicit knowledge or measurement of the causes of sex differences.”

As it stands, this manuscript is a valid contribution to the community. Still, the manuscript’s quality and clinical impact would benefit a lot if the authors could add and analyse the missing data to improve their predictive model’s clinical relevance.

Overall, I believe the authors have addressed the reviewers’ comments but with the caveat that a lot of the data is not accessible.

We appreciate this referee’s overall positive feedback. Eventually, we would like to emphasize that one of our main aims was to maximize the sample sizes of both cohorts to put findings on a firm basis – while we still needed to allow for clinical feasibility. Overall, we carefully accounted for all conceivably relevant inter-individual differences that we had access to: Sociodemographic characteristics (e.g., age), comorbidities (e.g., atrial fibrillation), markers of chronic brain health (e.g., white matter hyperintensity lesion load) and stroke subtypes (e.g., cardioembolic stroke genesis). Our core results replicated in an independent dataset and were reliably detectable in sensitivity analyses, that, for example, specifically addressed varying frequencies of cardioembolic stroke subtype and potentially associated differences in lesion patterns. All in all, we therefore feel confident that our presented sex-specific findings are valid. However, we acknowledge that even some of the largest currently available datasets do not comprise the level of detailed clinical information that we would like. This way, analyses of sex-specific lesion pattern-treatment effects may constitute exciting next research avenues.

References

1. Bonkhoff AK, Xu T, Nelson A, Gray R, Jha A, Cardoso J, Ourselin S, Rees G, Jäger HR, Nachev P. Reclassifying stroke lesion anatomy. *Under Review.*

2. Bonkhoff AK, Lim J-S, Bae H-J, Weaver NA, Kuijf HJ, Biesbroek JM, Rost NS, Bzdok D. Generative lesion pattern decomposition of cognitive impairment after stroke. *bioRxiv*. 2020;
3. Meyer L, Stracke CP, Jungi N, Wallocha M, Broocks G, Sporns PB, Maegerlein C, Dorn F, Zimmermann H, Naziri W. Thrombectomy for Primary Distal Posterior Cerebral Artery Occlusion Stroke: The TOPMOST Study. *JAMA neurology*. 2021;

REVIEWERS' COMMENTS

Reviewer #5 (Remarks to the Author):

I thank the authors for their careful consideration of my comments and for addressing them and where this was not possible due to missing data to caveat their findings and extend the limitations. I have no further requests for revisions.

Resubmission of revised manuscript: NCOMMS-20-40559-B

Reviewer #5 (Remarks to the Author):

I thank the authors for their careful consideration of my comments and for addressing them and where this was not possible due to missing data to caveat their findings and extend the limitations. I have no further requests for revisions.

Thank you, we appreciate this referee's positive evaluation of our revised manuscript greatly.